



# Defining aerosol layer height for UVAI interpretation using aerosol vertical distributions characterized by MERRA-2

Jiyunting Sun[1,2], J.Pepijn Veefkind[1,2], Peter van Velthoven[3], L.Gijsbert Tilstra[1], Julien Chimot[4], Swadhin Nanda[1,2], and Pieternel.F Levelt[1,2]

[1] Department of Satellite Observations, Royal Netherlands Meteorological Institute, De Bilt, 3731 GA, the Netherlands
[2] Department of Geoscience and Remote Sensing (GRS), Civil Engineering and Geosciences, Delft University of Technology, Delft, 2628 CD, the Netherlands
[3] Department of Weather & Climate Models, Royal Netherlands Meteorological Institute, De Bilt, 3731 GA, the Netherlands
[4] European Organization for the Exploitation of Meteorological Satellites (EUMETSAT), Darmstadt, 64295, Germany

*Correspondence to*: Jiyunting Sun (jiyunting.sun@knmi.nl)

**Abstract.** Aerosol vertical distributions are important for aerosol radiative forcing assessments and atmospheric remote sensing research. From our perspective, the aerosol layer height (ALH) is one of the major concerns in quantifying aerosol absorption from the ultra-violet aerosol index (UVAI). The UVAI has a global daily record since 1978, whereas a corresponding ALH data set is still limited. In this paper, we attempted to construct such an ALH data set from aerosol extinction profiles provided by the MERRA-2 aerosol reanalysis, meanwhile we evaluated them, together with several satellite ALH products in terms of the UVAI sensitivity to ALH. In the first part of this paper, we derived ALHs from the MERRA-2 aerosol profiles by four definitions. Through the sensitivity studies, we found that the definition of top boundary aerosol layer height ($H_{aer}^t$) is more robust to the changes in extinction profile properties than others. The spatial and temporal variation of $H_{aer}^t$ are also well associated with the major aerosol sources and the atmospheric dynamics. In the second part, we further evaluated the UVAI altitude dependence on the MERRA-2 ALH as well as several satellite ALH. Among all the satellite ALH products in this paper, the correlation between the TROPOMI oxygen ($O_2$) A-band ALH and UVAI, and that between the GOME-2 absorbing aerosol layer height (AAH) and UVAI are in agreement with our a-priori knowledge that the altitude dependence of UVAI increases with aerosol loadings. The correlation between the MERRA-2 $H_{aer}^t$ and UVAI also matches well with what we found from observational data sets. This implies the top boundary of the aerosol layer derived from MERRA-2 can be an alternative in case there is no observational ALH data available for quantitively aerosol absorption from UVAI and other UVAI-related applications.

## 1 Introduction

Atmospheric aerosols are small liquid or solid particles originating from natural or anthropogenic sources. Although initially emitted in the lower part of the atmosphere, aerosol particles can occasionally be transported across the tropopause and stay



in stratosphere for several months (Islam et al., 2017). Aerosol vertical distributions are affected by aerosol emissions and deposition processes, aerosol micro-physical properties, meteorological conditions, chemical processes, etc. Which one is the dominant factor determining the aerosol vertical distributions depends on aerosol species (Kipling et al., 2016). Aerosol vertical distributions are considered as one of the main contributors to the large uncertainties in assessments of aerosol radiative forcing (Boucher et al., 2013; Zhang et al., 2013; Pachauri et al., 2014).

Aerosol vertical distributions are also important for atmospheric remote sensing. For example, satellite retrievals of trace gases are sensitive to the relative location between trace gases and aerosols (Leitão et al., 2010; Shaiganfar et al., 2011; Ma et al., 2013; Kanaya et al., 2014; Liu et al., 2019). Aerosol optical properties retrievals also depend on the pre-assumed aerosol profiles in the radiative transfer simulations (Torres et al., 1998). From our perspective, interpretation of aerosol absorption from the ultra-violet aerosol index (UVAI) satellite records requires information on the aerosol vertical distributions (Sun et al., 2018; Sun et al., 2019), as UVAI is sensitive to the altitude. So far, UVAI has a four-decade global record, while a corresponding long-term global daily aerosol vertical distribution database is not yet available.

Aerosol vertical distributions are either described by aerosol profiles or columnar aerosol layer heights (ALH). Aerosol extinction profiles can be derived from lidar measured backscattered signals, such as the European Aerosol Research Lidar Network (EARLINET) (Pappalardo et al., 2014), the Micro-Pulse Lidar Network (MPLNET) (Welton et al., 2001), the Cloud-Aerosol Lidar with Orthogonal Polarization (CALIOP) onboard the Cloud-Aerosol Lidar and Infrared Pathfinder Satellite Observations (CALIPSO) (Winker et al., 2009), the Geoscience Laser Altimeter System (GLAS) onboard the Ice, Cloud and Elevation Satellite (ICESat) (Schutz et al., 2005), and the recently launched Atmospheric Dynamics Mission Aeolus (ADM-Aeolus) (Flamant et al., 2008). Although they provide great details in the vertical direction, lidar measured profiles are subjected to limited spatial and temporal coverage. Besides, the presence of clouds or optically dense aerosol layers may attenuate the lidar signal, resulting in large uncertainties or missing data in the measured profiles.

ALH is usually retrieved from passive sensor measurements. Many retrieval algorithms have been developed. Multiple angular measurements can determine elevated aerosol plume height by stereo photogrammetry, e.g. the Multi-angle Imaging Spectroradiometer (MISR) onboard Terra (Nelson et al., 2013). The POLarization and Directionality of the Earth's Reflectance (POLDER) onboard PARASOL utilizes the distinct polarization difference between air molecules and aerosol particles in the near-UV spectrum (Dubovik et al., 2011). ALH derived by spectrum fitting over oxygen ($O_2$) absorption A-band is more commonly applied. Representative instruments includes POLDER, the MEdium Resolution Imaging Spectrometer (MERIS, Duforêt et al., 2007; Dubuisson et al., 2009) and the SCanning Imaging Absorption SpectroMeter for Atmospheric CHartographY (SCIAMACHY) onboard ENVISAT (Sanghavi et al., 2012), the Global Ozone Monitoring Experiment–2 (GOME-2) onboard Metop-B (Sanders et al., 2015), the TROPOspheric Monitoring Instrument (TROPOMI) onboard Sentinel-5P (Sanders A.F.J. and de Haan J.F., 2016), and the Earth Polychromatic Imaging Camera (EPIC) onboard DSCOVR (Xu et al., 2017). Chimot et al. (2017) also attempted to retrieve ALH from $O_2$-$O_2$ absorption band (477 nm) for the Ozone Monitoring Instrument onboard Aura (Chimot et al., 2017). Recently, the absorbing aerosol layer height (AAH) has become an official product of the GOME-2 instrument that retrieves height of absorbing



aerosol layers (Tilstra et al., 2019). Instruments equipped with thermal infrared (thermal IR) band, such as the Atmospheric InfraRed Sounder (AIRS) onboard Aqua and the Infrared Atmospheric Sounding Interferometer (IASI) onboard Metop-A, can retrieve height information for dust because the thermal IR band is highly sensitive to dust aerosols (Pierangelo et al.,

2010; Vandenbussche et al., 2013). For a detailed review of ALH retrievals from observations, one can refer to Islam et al. (2017). Although extending the spatial coverage of lidar measurements, ALH retrievals from passive sensor measurements are only applicable under certain conditions, e.g. elevated aerosol layers, over dark surfaces, in the absence of clouds, etc. Moreover, assumptions on aerosol optical properties, aerosol profiles and ALH definitions in the forward simulations and backward retrievals vary with algorithms and spectral bands.

ALH can also be calculated from aerosol extinction profiles. This is necessary, for example, when one wants to compare columnar ALH with three-dimensional aerosol profiles. However, as we will introduce in Section 2, there exist various ALH definitions. From our perspective, which ALH definition can be used to interpret aerosol absorption from UVAI is the main research question of this paper. To answer this question, we will study the properties of various ALH definitions and investigate their relationship with UVAI. Considering the limited horizontal coverage and the complex vertical structure of

the lidar measured aerosol profiles, we use aerosol profiles provided by a chemistry transport model (CTM), i.e. the Modern-Era Retrospective Analysis for Research and Applications, version 2 (MERRA-2) instead. The aerosol fields of MERRA-2 are available since 1980 and generally in good agreement with independent measurements, which are well-documented in Randles et al. (2017) and Buchard et al. (2017).

Once obtaining the MERRA-2 ALHs, we will evaluate them, together with several satellite ALH products, in terms of UVAI

altitude dependence on them. We use the UVAI altitude dependence as the evaluation criterion for two reasons. Firstly, direct comparisons between different ALHs are not very meaningful because they represent the different aspects of the aerosol vertical distributions (Torres et al., 2013); secondly, our purpose is to find an ALH data set for interpreting aerosol absorption from UVAI. The result of this paper will tell which satellite ALH product is most suitable in terms of analysis aerosol absorption properties and build up a long-term global ALH data set characterized by the MERRA-2 aerosol fields for

future UVAI-related applications.

This paper is constructed as follows: section 2 starts with introduction of the MERRA-2 aerosol reanalysis data set and different ALH definitions, followed by an analysis of the derived MERRA-2 ALH quantities. Section 3 introduces the selected satellite ALH products to evaluate the performances of the derived MERRA-2 ALHs. Section 4 analyzes the MERRA-2 ALHs and their relationship with UVAI, followed by a comparison between MERRA-2 derived ALH and

satellite ALH products. Section 5 summaries the conclusions and gives outlooks for future applications.

## 2 MERRA-2 aerosol layer heights

This section starts with a brief introduction and validation of MERRA-2 aerosol reanalysis data. Then, we propose four candidate ALH definitions and apply them to the MERRA-2 aerosol profiles. Sensitivity studies of the derived ALHs are





performed to examine their robustness to the changes in aerosol profiles. Lastly, we analyze the spatial and temporal

distribution of the derived MERRA-2 ALHs.

**2.1 MERRA-2 aerosol reanalysis**

MERRA-2 is the latest modern satellite era (1980 onwards) atmospheric reanalysis from NASA Global Modeling and

Assimilation Office (GMAO) (Buchard et al., 2017). MERRA-2 employs the Goddard Earth Observing System, version 5

Earth system model (GEOS-5) (Molod et al., 2015) and the three-dimensional variational data assimilation (3DVar)

Gridpoint Statistical Interpolation analysis system (GSI) (Kleist et al., 2009). The GEOS-5 is coupled to the Goddard

Chemistry Aerosol Radiation and Transport model (GOCART) aerosol module (Chin et al., 2002; Colarco et al., 2010). The

model resolution is $0.5° \times 0.625°$ latitude by longitude with 72 hybrid-eta layers from the surface up to $0.01\ hPa$.

MERRA-2 assimilates multiple observational AOD data sets. The radiances of Moderate Resolution Imaging

Spectroradiometer (MODIS) and the Advanced Very High Resolution Radiometer (AVHRR) are translated into the

AERONET (AErosol RObotic NETwork)-calibrated AOD via the Neural Net Retrieval (NNR) algorithm. The system also

assimilates MISR AOD over bright surfaces (surface albedo > 0.15) to include desert regions and ground based AERONET

measurements. MERRA-2 aerosol assimilation and the total columnar AOD evaluation are well-documented in (Randles et

al., 2017). They elaborate that the MERRA-2 AOD constrained by observations better matches independent measurements.

Improved agreement is also found for aerosol optical properties and aerosol vertical distributions (Buchard et al., 2017).

In this paper, we employ the MERRA-2 3-hourly instantaneous aerosol mass mixing ratio (MERRA-2 inst3_3d_aer_Nv,

10.5067/LTVB4GPCOTK2 , last access: 7 June 2019). The selected period is from 2006-01-01 to 2016-12-31. We use the

mean value between 12:00 and 15:00 local time in order to be consistent with the satellite observations that are of most

interests to us (OMI and TROPOMI, both have an overpass time at around 13:30 of local time, Levelt et al., 2006; Veefkind

et al., 2015). MERRA-2 provides mass mixing ratio profiles for 15 aerosol sub-species, including dust (5 noninteracting size

bins), sea salt (5 noninteracting size bins), hydrophobic and hydrophilic black and organic carbon (BC and OC), and sulfate

($SO_4$). A conversion from mass concentrations ($c$, unit: kg kg$^{-1}$) to extinction coefficients ($\beta$, unit: km$^{-1}$) is necessary in order

to allow comparisons with satellite measurements. This procedure is explicitly described in Appendix A. From here

onwards, the term 'MERRA-2 aerosol profiles' exclusively indicates the 'MERRA-2 aerosol extinction coefficient profiles'

in this paper.

Aerosol profiles simulated by CTMs may have an order of magnitude difference (Koffi et al., 2012; Kipling et al., 2016). To

prove that the MERRA-2 aerosol fields are realistic, we validated the converted MERRA-2 extinction profiles and the

columnar AOD with the CALIOP monthly climatology during the selected period in Appendix B. Generally, the aerosol

fields of MERRA-2 are in good agreement with CALIOP, with respect to both the spatial distribution and the seasonal

variation. One can refer to Appendix B for more details.


## 2.2 Definitions of aerosol layer height

There are various methods to derive an ALH value from a given aerosol extinction profile. One can calculate the aerosol effective heights, for example, the aerosol mean height weighted by the aerosol properties (aerosol properties-weighted mean height), or the aerosol scale height at which the aerosol profile or the cumulative profile passes a pre-determined threshold. One can also detect the geometric boundary or center (aerosol geometric height), the so-called aerosol layer real height. In this section, we introduce the above ALH definitions in detail and apply them to the MERRA-2 aerosol extinction profiles. Table 1 lists the above representative ALH definitions. Note that all ALHs are calculated from full profiles from the surface to the top of atmosphere (TOA) and they are relative to the terrain height by default unless it is mentioned precisely.

### 2.2.1    Aerosol optical properties-weighted mean height

Given an aerosol extinction coefficient profile ($\beta(z)$) with $n$ layers, a common way to derive the ALH is calculating the mean height weighted by the extinction coefficient in each atmospheric height interval (Eq.(1), Koffi et al., 2012; Chimot et al., 2018; Kylling et al., 2018; Liu et al., 2019) or by the AOD in each atmospheric height interval (Eq.(2), Wu et al., 2016).

$$H_{aer}^{\beta} = \frac{\sum_{i=1}^{n} \beta(z_i) \cdot z_i}{\sum_{i=1}^{n} \beta(z_i)} \tag{1}$$

$$H_{aer}^{\tau} = \frac{\sum_{i=1}^{n} \beta(z_i) \cdot dz_i \cdot z_i}{\sum_{i=1}^{n} \beta(z_i) \cdot dz_i} \tag{2}$$

, where $\beta(z_i)$ and $dz_i$ are the extinction coefficient and the geometric thickness of each atmospheric height interval $i$. The superscript $\beta$ and $\tau$ of $H_{aer}$ indicate the averaging weight is extinction coefficients or AOD. If the atmospheric layers are evenly gridded in vertical direction, then the $H_{aer}^{\beta}$ and $H_{aer}^{\tau}$ will give the same result.

### 2.2.2    Aerosol scale height

The aerosol scale height used to assume that the extinction profiles exponentially decay with altitude ((Hayasaka et al., 2007; He et al., 2008; Chu et al., 2009). This assumption restricts the application of the scale height to the condition when the extinction peaks near the surface. Alternatively, a more generalized scale height definition is proposed (Turner et al., 2001; Hayasaka et al., 2007; Leon et al., 2009; Yu et al., 2010).

$$\int_{0}^{H_{aer}^{63}} \beta(z)dz = \sum_{i=1}^{m} \beta(z_i) \cdot dz_i = \tau(1 - e^{-1}) \approx 0.63\tau \tag{3}$$

, where $\beta(z_i)$ and $dz_i$ are the extinction coefficient and the geometric thickness of each atmospheric height interval $i$, $m$ is the number of layers up to $H_{aer}^{63}$ and $\tau$ is the total columnar AOD.





### 2.2.3    Aerosol geometric height

The above ALH definitions are 'effective' heights where aerosol loading should be placed to be representative of the aerosol radiative properties, while the aerosol geometric height describes the 'real' aerosol layer location (Kylling et al., 2018). The

geometric height commonly presented by the central height or the boundary of an aerosol layer. The central height is usually the mean of an aerosol top and bottom boundary (Kylling et al., 2018). The bottom boundary is more difficult to determine than the top boundary if there are multiple aerosol layers. Here, we only focus on how to define the top boundary of the highest aerosol layer ($H_{aer}^t$).

For a box-shape profile, $H_{aer}^t$ is explicitly indicated by a clear sharp decrease of the extinction coefficient in the transition

layer. For other profile types, there is no uniform method to determine $H_{aer}^t$. Welton et al. (2002) found the top boundary if the lidar signal strength is greater than the Rayleigh signal by a predetermined threshold setting. The mean signal over the next 500 m is also checked in order to avoid effect from noise. Leon et al. (2009) detected the top layer boundary by retaining the first altitude below TOA at which the signal is 3 times of the standard deviation larger than the average in the reference altitude (6.5 to 7 km). CALIOP employs a much more comprehensive layer detection algorithm (SIBYL, Vaughan

et al., 2009), where the magnitude of the threshold is adapted according to the characteristics of the signal (Winker et al., 2009).

For an extinction profile provided by MERRA-2, we attempt to find the top border of an aerosol layer with help of the extinction coefficient lapse rate ($\gamma_{ext}$, unit: km$^{-2}$), which is defined as:

$$\gamma_{ext}(z) = -\frac{d\beta(z)}{dz}$$

(4)

, where $d\beta(z)$ is the extinction coefficient difference between two continuous layers and $dz$ is the atmospheric interval

geometric thickness. The concept of $\gamma_{ext}$ is proposed by Huang et al. (2017) to explore the relationship between atmospheric stability and aerosol vertical distributions. The stable meteorological conditions lead to a large positive $\gamma_{ext}$, while the elevated aerosol layers result in a negative value. Given an aerosol extinction profile, we search upwards from the surface and retain the first height at which the magnitude of $\gamma_{ext}$ above this height is always smaller than a certain value. We set up a sensitivity study in Appendix C and find the threshold of 0.01 km$^{-2}$ is most suitable to determine the aerosol layer top

boundary.

### 2.3 MERRA-2 aerosol layer height sensitivity studies

The derived ALHs using above methods may depend on the attributes of given aerosol extinction profiles. For instance, whether an extinction profile is evenly gridded in vertical direction, to which altitude an extinction profile extends and what is the minimum level of an extinction coefficient is used (i.e. background values in a model or detection limit of an

instrument) may influence the value of derived ALHs. Therefore, we explore the sensitivities of ALHs to the above aspects.





The sensitivity studies are based on a subset of the daily MERRA-2 aerosol profiles during the period from 2006-01-01 to
2016-12-31. For each day, we randomly select 100 profiles, a total of 401800 profiles are selected. For each profile, we
modify the profile by changing its vertical grid size, truncating it to a certain height (the profile top height), or resetting the
minimum extinction coefficient. Then we compare the ALHs derived from the original MERRA-2 profiles and that derived

from the modified profiles by making difference between them (the latter minus the former).

Fig.1 shows the statistics of each ALH quantity. The original vertical grid in MERRA-2 is irregular, i.e. denser grids in the
lower part of the atmosphere and coarser grids in the upper. This has a significant influence on $H_{aer}^{\beta}$, though the effect of
vertical resolution is limited as its calculation is independent to the grid size of each layer. The bias of other ALH definitions
is small if the vertical resolution is less than 0.5 km. Compared with $H_{aer}^{t}$, the other ALH definitions are more sensitive to

the top height of the profile, as their calculation depends on the full profile. If the profile only contains information within
the troposphere (say less than 15 km), the derived ALHs may be significantly underestimated compared with that derived
from the full profile. The influence of profile top height is negligible above 20 km as there exists little aerosol. The minimum
extinction coefficient has an obvious effect on the derived ALHs only if it is larger than 0.001 km⁻¹. Nevertheless, $H_{aer}^{t}$ is
almost insensitive to the minimum extinction.

The sensitivity studies give us several implications which may useful when one wants to derive an ALH from a given
extinction profile: whether a profile is evenly gridded in vertical direction or not may lead to significant difference in $H_{aer}^{\beta}$;
the profile vertical resolution is suggested to be better than 0.5 km; obtaining full profiles from the surface to the top of the
atmosphere is generally not necessary, but it is suggested the profile top height should not be lower than 20 km; the
minimum extinction coefficient matters if it is over 0.001 km⁻¹. Generally, among all ALH definitions, $H_{aer}^{t}$ seems to be the

most robust that is not vary significantly due to the changes in the given aerosol profile.

**2.4 Spatial and temporal distribution of MERRA-2 aerosol layer heights**

Fig.2 shows the global seasonal climatology maps of MERRA-2 ALHs averaged over the selected period. The magnitude of
$H_{aer}^{\beta}$ is lowest, with most values less than 2 km. This is because the aerosol extinction is usually higher at the lower part of
the atmosphere and the maximum extinction coefficient often appears at or near the ground, as we can tell from the zonal

extinction profile climatology (Fig.B1) in Appendix B. Besides, $H_{aer}^{\beta}$ does not account for the varying thickness of each
atmospheric height intervals. The denser grid in the lower part of the atmosphere gives more weights to the lower altitudes.
On the contrary, $H_{aer}^{\tau}$ and $H_{aer}^{63}$ are calculated based on AOD profiles. Consequently, the ALH values are no longer trapped
in the lower part of the atmosphere as $H_{aer}^{\beta}$. One should note that if the vertical grid is equidistant, $H_{aer}^{\beta}$ and $H_{aer}^{\tau}$ will give
the same result.

The spatial pattern of effective heights ($H_{aer}^{\beta}$, $H_{aer}^{\tau}$ and $H_{aer}^{63}$) are similar, despite that the spatial variability of $H_{aer}^{\beta}$ is much
weaker. Compared with high aerosol loading regions (presented by AOD seasonal climatology in Appendix B Fig.B2), for



example, the dust belt and biomass burning regions, these three quantities are significantly higher in the rest of the world, particularly in high-latitude clean regions. The seasonal variations also show that the effective heights are more variable over low AOD regions than aerosol source regions. By contrast, the spatial-temporal variation of $H_{aer}^t$ is better associated with

AOD. One can easily recognize the seasonal aerosol sources from the spatial and temporal variation of $H_{aer}^t$, e.g. the biomass burning regions in the central Africa during winter, the Sahara dust and its outflows over the Northern Atlantic during summer, etc.

Fig.3 shows the zonal average and standard deviation of MERRA-2 ALHs against the corresponding zonal average of MERRA-2 extinction coefficient profiles for the selected period. The behavior of $H_{aer}^t$ in general follows the contours of the

extinction coefficients. The magnitude and variability of $H_{aer}^t$ is lower in the Southern Hemisphere, with the lowest level located at the South Pole, because the major aerosol sources are located in the Northern Hemisphere, and atmospheric dynamic is more active at tropic and subtropical regions. The effective heights ($H_{aer}^\beta$, $H_{aer}^\tau$ and $H_{aer}^{63}$) behave in a similar way, though the magnitude of $H_{aer}^\beta$ is much lower. These three presents significantly higher ALH over polar areas where the aerosol loading is little. Whereas in aerosol source regions, the ALH values are closer to the surface due to the more

importance are given in the lower part of the atmosphere. The effective heights, particularly $H_{aer}^\tau$ and $H_{aer}^{63}$, also show significant variabilities over low AOD regions.

Compared with $H_{aer}^t$, it is difficult to recognize aerosol sources from both the spatial maps and the zonal profiles of $H_{aer}^\beta$, $H_{aer}^\tau$ and $H_{aer}^{63}$. The reason is that they are sensitive to where most photons extinct (the location of the peak extinction layers), how prominent those layers compared to the rest (the magnitude of the peak extinction layers), and how fast the

extinction coefficient decay ($\gamma_{ext}$). Here, we take four representative extinction coefficient profiles as examples to explain the behavior of these ALH definitions. Fig.4 shows the representative extinction profiles (black lines). They are obtained from the average extinction profiles during the selected period for East China, North Africa, Antarctica and South Africa. For each region, we also provide the standard deviation to show the variation of the mean profiles (gray bars), and $\gamma_{ext}$ to show how fast the extinction profiles changes with height (blue lines).

Although both the mean profile of East China (Fig.4a) and North Africa (Fig.4b) monotonically decrease with altitude, the magnitude of $\gamma_{ext}$ of North Africa is much smaller. The extinction coefficient of East China is no longer significant above 3 km due to the fast decay, whereas at the same altitude, the extinction coefficient of North Africa still plays an important role in ALH calculations. As a result, the derived ALHs for North Africa are slightly higher than that of East China. The mean profile of Antarctica also decays with altitude (Fig.4c), however the peak extinction is only slightly higher than extinction in

other layers. As the weights given to each atmospheric layer are not significantly different in this case, the role of extinction coefficients in ALH calculations is not as important as that in East China and North Africa. Instead, the three effective heights are dominated by the atmospheric interval geometric thickness. Consequently, the derived ALHs, particularly $H_{aer}^\tau$ and $H_{aer}^{63}$, may indicate an altitude where there is little aerosol loading. Differing from other cases, the mean profile of South Africa (Fig.4d) has a weak inversion layer (i.e. negative $\gamma_{ext}$) and peaks at around 1.5 km. With less importance given to the





extinction coefficients below this altitude, the derived $H_{aer}^{\beta}$, $H_{aer}^{\tau}$ and $H_{aer}^{63}$ are higher than other cases (Fig.4a-c) where their peak extinctions layers are at the surface.

## 3 Satellite aerosol layer height products

In this section, we focus on the introduction and pre-processing of several satellite ALH products which also have corresponding UVAI available. As listed in Table 2, the candidate products are the ALH reported in the OMAERUV, the

OMI $O_2$-$O_2$ neural network ALH, the TROPOMI $O_2$ A-band ALH and the GOME-2 Absorbing aerosol layer height. It is noted that the OMAERUV ALH is not an independent product retrieved based on physical processes, but a best guess based on satellite and CTM climatology. The reason we include it in this paper is because that the OMAERUV ALH is designed to retrieve accurate aerosol properties of absorbing aerosols in the UV channel (Torres et al., 2013), and it has a long-term global record since 2006.

### 3.1 OMAERUV aerosol layer height

The ALH provided by the official OMI/Aura level 2 OMAERUV product (http://dx.doi.org/10.5067/Aura/OMI/DATA2004, last access: 12 August 2019) is a best guess to retrieve accurate aerosol properties of absorbing aerosols in the UV channel (Torres et al., 2013), which is the main reason we include this ALH data set in this paper. The ALH is either given by the CALIOP climatology, a CTM provided climatology (for dust), or a-priori assumptions (for carbonaceous and sulphate

aerosols) according to aerosol types and geolocations if the CALIOP entry is not available. The aerosol types in OMAERUV are determined by the corresponding UVAI, the carbon monoxide data from the Atmospheric Infrared Sounder (AIRS) and the scene type from the International Geosphere/Biosphere Programme (IGBP) database. One can refer to Torres et al. (2013) for more detailed information.

The OMAERUV data used in this paper are from the same period as MERRA-2, i.e. 2006-01-01 to 2016-12-31. Satellite

pixels with solar zenith angle (SZA) larger than 70°, or contaminated by clouds (cloud fraction larger than 0.3), sun-glint (glint angle larger than 20° over water) or the so-called row anomaly (XTrackQualityFlags is not 0) are removed before analysis. Fig.5 a and 5b present the distribution of the OMAERUV ALH and its corresponding UVAI. Note that the parameters in Fig.5 is normalized by the maximum value in each subplot. The highest UVAI values show up at North Africa and its dust outflows over the north Atlantic Ocean, the Arabic peninsula and the Gobi Desert. The smoke plumes generated

by the biomass burning events in the central and southern Africa are also visible. Apart from remote oceans, the spatial distribution of ALH is positively associated with UVAI. The extreme high ALH over oceans (Fig.5a) may be caused by many factors. For instance, the row anomalies and sun-glint pixels that cannot be detect by the quality flag may lead to large UVAI, which further affects the classification of aerosol types and ALH determinations; the unrealistic a-priori assumptions of ALH when no climatological entry exists; or the high sensitivity to outliers as the number of observations is small over





remote oceans, etc. Thus, to ensure the quality of OMAERUV ALH in the further analysis, we employ the ground network
       AERONET data as an additional quality control, which will be described later in Section 4.2.

**3.2  OMI O2-O2 neural network aerosol layer height**

The $O_2$-$O_2$ ALH retrieval algorithm is based on the OMI slant column density (SCD) of $O_2$-$O_2$ at 477 nm (Chimot et al.,
2017). The principle behind is to detect how aerosols affect the average length of $O_2$-$O_2$ absorption light path. An aerosol
layer located at higher altitude suggests less photons reach the lower part of the atmosphere beneath the aerosol layer
       compared with an aerosol-free case, i.e. a larger shielding effect reducing the $O_2$-$O_2$ absorption.
       The $O_2$-$O_2$ ALH retrieval is realized by a neural network (NN) algorithm (Chimot et al., 2017). The training process is fully
       based on radiative transfer simulations for two types of Henyey-Greenstein aerosols (Ångström Exponent = 1.5 and
       asymmetry factor = 0.7) with different single scattering albedo values at 550 nm (SSA = 0.90 and 0.95) under cloud-free
scenes. The aerosol profile is parameterized as a box-shape profile with constant geometric thickness of 1 km. The input
       features of NN consist of satellite measurement geometries, surface pressure and surface albedo, $O_2$-$O_2$ SCD at 477 nm and
       AOD at 550 nm. For more information on the configuration of the $O_2$-$O_2$ ALH neural network retrieval algorithm, one can
       refer to Chimot et al. (2017). The accuracy of the $O_2$-$O_2$ ALH is between 0.5 and 1 km (Chimot et al., 2017). Chimot et al.
       (2018) compared the $O_2$-$O_2$ ALH with LIVAS climatology for the northeast Asia during 2005-2007 and find a maximum
difference less than 800 m. They also find good agreements between the $O_2$-$O_2$ ALH and the CALIOP level 2 data for cases
       with different aerosol types.
       The $O_2$-$O_2$ ALH global data set used in this study is available only for year 2006 (the data is not publicly assessible, last
       access: 15 July 2019). It is trained by the MODIS AOD at 550 nm of the combined dark target and deep blue products, and
       the $O_2$-$O_2$ SCD originally from the OMICLDO2 product. As the algorithm is exclusively trained under cloud-free scenes, we
filter out pixels with MODIS geometric cloud fraction larger than 0.02 or OMI cloud fraction larger than 0.1. Besides, pixels
       with low aerosol loading (AOD smaller than 0.5) are abandoned as the aerosol shielding effect on $O_2$-$O_2$ is negligible
       (Chimot et al., 2018). To avoid the large errors triggered by NN extrapolation, i.e. samples with feature values outside the
       range in the training data set, we only retain samples satisfying the following criteria: surface albedo is less than 0.1, surface
       height is less than 400 m, SZA is between 9° and 65°, the viewing zenith angle (VZA) is between 0° and 45° and retrieved
aerosol layer pressure is between 150 and 975 hPa. More information can refer to the Table 1 in Chimot et al. (2017).
       Fig.5d and 5g present the distribution of the $O_2$-$O_2$ ALH trained for aerosol SSA of 0.90 and 0.95, respectively. The UVAI in
       Fig.5e and 5h is the co-located UVAI provided by the OMAERUV products. The $O_2$-$O_2$ ALH is unevenly distributed with
       majority of them located in the central Africa, North India and East China, while little retrievals over desert regions due to
       the bright surface. The one-year data also reveals the biomass burning events in North America, South America and
Australia. The $O_2$-$O_2$ ALH trained with aerosol SSA of 0.95 is generally higher than that trained by SSA of 0.90, since the
       measured $O_2$-$O_2$ absorption is lower than expected (Chimot et al., 2017).



### 3.3 TROPOMI O2 A-band aerosol layer height

The TROPOMI $O_2$ A-band ALH is developed to detect vertically localized aerosol layers in the free troposphere under clean sky, i.e. dust storms, biomass burning or volcanic plumes (Sanders A.F.J. and de Haan J.F., 2016). The ALH is based on the

$O_2$ absorption in the near-infrared (758 – 770 nm). Similar to OMI $O_2$-$O_2$ absorption at 477 nm, a higher $O_2$ absorption indicates a longer absorption light path as the aerosol layer is close to the surface (Sanders et al., 2015).

To accelerate the computational efficiency in operation, a NN algorithm is implemented to replace the line-by-line radiative transfer method during the retrieval procedure and to generate the official TROPOMI ALH product (Nanda et al., 2019a). The training process is also based on radiative transfer simulations but only for a fixed Henyey-Greenstein aerosol type (SSA

at 550 nm = 0.95, Ångström Exponent = 0 and asymmetry factor = 0.7). The aerosol layer is assumed to be distributed in a homogeneous layer with constant thickness of 50 hPa. Other input features of the radiative transfer simulations are satellite measurement geometry, aerosol optical properties, meteorological parameters and surface conditions. For more detailed information on this algorithm one can refer to Nanda et al. (2019). The validation study shows that TROPOMI ALH is generally in good agreement with collocated CALIOP level 2 data. The CALIOP ALH is generally higher than TROPOMI

(1 km higher over ocean and 2.4 km higher over land) as CALIOP is more sensitive to the top of the aerosol layer where most of the signal comes from (Nanda et al., 2019, AMTD).

In this study, we collect the TROPOMI ALH level 2 offline data from 2018-11-03 to 2019-08-31 (https://s5phub.copernicus.eu/dhus/#/home , last access: 18 November 2019). The TROPOMI UVAI and FRESCO cloud fraction are enclosed in this product. The data product also retrieves AOD as a diagnostic tool indicating influence of bright

surfaces and undetected clouds. Similar to the pre-processing of the OMI $O_2$-$O_2$ ALH, to avoid NN extrapolation due to scenes not included in the training process, samples are kept only if they satisfy the following criteria: AOD between 0.05 and 5, SZA between 8.2° and 70°, ALP between 75 and 1000 hPa, surface pressure between 520 and 1048.5 hPa and surface albedo smaller than 0.7. For more details, one can refer to Table 2 in Nanda et al. (2019). To exclude out ALH quality, we only retain pixels with only successful retrievals (Processing Quality Flags = 0) and full quality data (qa_value = 1). This

procedure automatically excludes pixels affected sun-glints, clouds, bright surfaces (snow and ice) and UVAI (calculated by 354-388 nm wavelength pair) smaller than 1. Besides, as aerosols transported from a certain distance are assumed smoothly distributed, to avoid sub-pixel cloud contamination, a local standard deviation for a pixel and its nearest surrounding 8 pixels is calculated. A pixel is excluded if its local standard deviation of ALH is higher than 0.2 km or its standard deviation of AOD at 758 nm is higher than 10.

Fig.5j and 5k presents the distribution of TROPOMI ALH and the corresponding level 2 TROPOMI UVAI used in the retrieval. The majority of the data are located at outflows of Sahara dust and the central Africa biomass burning. The less-than-one-year data also reveals fire events happened in California, Canada, South America, Russia and Australia.





### 3.4 GOME-2 absorbing aerosol layer height

The GOME-2 AAH is derived based on the GOME-2 UVAI product (Tilstra et al., 2010) and the FRESCO cloud product
(Wang et al., 2008). FRESCO retrieves effective cloud pressure and cloud fraction using the reflectance of the $O_2$ A-band at
760 nm. Since this wavelength is suitable to retrieve ALH for cloud-free cases (Boesche et al., 2009; Dubuisson et al., 2009;
Sanders et al., 2015) and aerosols are treated in the same way as clouds in FRESCO, Wang et al. (2012) attempted to derive
and interpret information on aerosol layer pressure from the FRESCO cloud product. Their findings led to the operational
GOME-2 AAH algorithm (Tilstra et al., 2019). The algorithm retrieves ALH only for pixels with UVAI larger than 2 with
two methods. In the first approach, cloud pressure is retrieved along with effective cloud fraction by assuming a constant
cloud albedo of 0.8. Another approach retrieves scene pressure and scene albedo if the cloud/aerosol layer is assumed to
cover the entire scene. If the retrieved cloud fraction is below 0.25 or higher than 0.75, the retrieved cloud pressure can better
represent the aerosol layer pressure. In other situations, the best estimate of ALH is the higher value between cloud pressure
and scene pressure. For more detailed algorithm description one can refer to Tilstra et al. (2019).
We collect the AAH provided by the GOME-2 on-board the Metop-A/B/C from 2018-08-01 to 2019-10-31 (the data are not
publicly accessible yet, last access: 24 November 2019). Pixels with SZA larger than 70°, or those affected by sun-glint or
solar eclipse events are abandoned (AAH_Error_Flag = 0). The GOME-2 AAH retrieves ALH only for pixels with UVAI
larger than 2 (Tilstra et al., 2019). Besides, unconverging pixels with AAH set to be 15 km are also excluded.
Fig.5m and 5n presents the distribution of GOME-2 AAH and the corresponding level 2 GOME-2 UVAI (calculated by 340-
380 nm wavelength pair) product. The distribution of AAH is similar to that of the TROPOMI ALH as they are available
during a similar period and they are retrieved in the same band. The fire events reflected by TROPOMI ALH also appears in
the GOME-2 AAH map. While the TROPOMI ALH has more observations of dust and smoke outflows over the Atlantic
Ocean, the GOME-2 AAH has better availability over desert regions and remote oceans as its retrieval has no constraint on
surface albedo and cloud fraction.

### 360 4   Comparison between MERRA-2 ALHs and satellite ALH products

In previous sections, we have obtained four MERRA-2 ALHs using different definitions and four pre-processed satellite
ALH products. A direct comparison between above ALH data sets is not very meaningful as these definitions represent
different aspects of the aerosol vertical distribution (Torres et al., 2013). Furthermore, the aerosol vertical profile
parameterizations, the retrieval principles, the measurement techniques are not the same. Instead, we focus on the
comparison in terms of the relationship between ALH and UVAI. This section starts with introducing UVAI and its
dependence on ALH. Next, as CMTs do not have corresponding UVAI fields, we assign the OMAERUV UVAI data set to
the MERRA-2 ALHs with quality assurance using independent AERONET records. Lastly, we discuss the performances of
the MERRA-2 ALHs and satellite ALH products in terms of UVAI dependence on them.



### 4.1 UVAI and its dependence on ALH

UVAI is a long-term record of aerosol absorption since 1978 (Herman et al., 1997). It detects the change of the radiance or reflectance contrast between two UV channels ($\lambda_1$ and $\lambda_2$) due to the presence of absorbing aerosols:

$$UVAI = -100 log_{10}\left[\left(\frac{R_{\lambda 1}}{R_{\lambda 2}}\right)^{obs} - \left(\frac{R_{\lambda 1}}{R_{\lambda 2}}\right)^{Ray}\right] \tag{5}$$

, where *obs* and *Ray* indicate measured and simulated for a Rayleigh atmosphere, respectively. $R_{\lambda 1}^{Ray}$ is simulated at the surface albedo that satisfies $R_{\lambda 2}^{obs} = R_{\lambda 2}^{Ray}$. A positive UVAI value represents the presence of absorbing aerosols while a negative value indicates the non-absorbing components.

The UVAI dependence on ALH is well-studied by radiative transfer simulations, no matter which aerosol profile assumption is made (Herman et al., 1997; Torres et al., 1998; de Graaf et al., 2005; Sun et al., 2018). Here we set up a simple sensitivity study to review how UVAI response to ALH using the radiative transfer model Determining Instrument Specifications and Analysing Methods for Atmospheric Retrieval (DISAMAR) (de Haan, 2011). In the forward radiative transfer simulations, the aerosols are characterized by the Henyey-Greenstein (HG) phase function with a constant asymmetry factor of 0.7 and an

Ångström Exponent of 1. The aerosol layer is parameterized as a box-shape profile with a constant depth of 50 hPa. The input AOD at 550 nm varies between 0.01 and 3 and the input ALH varies between 0.5 and 12 km. Other inputs and detailed settings for the radiative transfer simulations are provided in Appendix D. We design several scenarios by choosing either absorbing (SSA = 0.90) or scattering (SSA = 0.99) aerosols over either a dark (surface albedo = 0.05) or bright surface (surface albedo = 0.3).

Fig.6 shows the results. UVAI increases with ALH due to more molecular radiation coming from below is absorbed by aerosols. The higher aerosol loadings, the stronger the UVAI dependence on ALH (Fig.6a). However, the dependence becomes weaker over brighter surfaces, particularly under low aerosol loading (Fig.6b). On the other hand, little altitude dependence is found for scattering aerosols (Fig.6c). The above features of UVAI-ALH relationship can be used to validate the performances of MERRA-2 ALHs and satellite ALH products. As the influence of surface albedo is negligible compared

with that of AOD, in the following analysis, we only focus on the UVAI dependence to ALH under different aerosol loadings.

### 4.2 Assignment of the OMAERUV UVAI to the MERRA-2 ALHs

The UVAI data we use to examine the MERRA-2 ALHs is provided by the OMI/Aura level 2 OMAERUV product (http://dx.doi.org/10.5067/Aura/OMI/DATA2004, last access: 12 August 2019) from 2006-01-01 to 2016-12-31. We first

resample the MERRA-2 ALHs onto the pre-processed OMAERUV data as described in Section 3.1. To ensure the consistency between the two independent data sets, we employ the AERONET version 3 level 1.5 direct-sun and inversion products (https://aeronet.gsfc.nasa.gov , last access: 18 November 2019, Holben et al., 1998) for the quality control. The co-location criteria for the three data sets follow previous studies (Remer et al., 2002; Torres et al., 2002; Bréon et al., 2011;



Jethva et al., 2014; Lacagnina et al., 2015): given an AERONET record, a time window ($\pm30$ minute for AOD and $\pm3$ hour

for SSA) and a spatial threshold ($\leq50$ km) is applied to the OMAERUV-MERRA-2 data set. The joint satellite-model pixels

are averaged if they both pass the quality control on both AOD and SSA (Dubovik et al., 2000; Remer, 2005):

(1)  $\Delta AOD_\lambda = 0.05 + 0.15 \times AOD_{AERONET,\lambda}$  over land

(2)  $\Delta AOD_\lambda = 0.03 + 0.05 \times AOD_{AERONET,\lambda}$  over ocean

(3)  $\Delta SSA_\lambda \leq 0.03$

, where $\lambda$ is the wavelength which is 500 nm for OMAERUV and 550 nm for MERRA-2. For AERONET sites do not report

AOD at 500 nm or 550 nm, the $AOD_{500}$ or $AOD_{550}$ is estimated by the Ångström Exponent and AOD at other channels. The

$SSA_{500}$ or $SSA_{550}$ is linearly interpolated by the nearest values (usually 440 nm and 675 nm, but the exact wavelength may

slightly vary from site to site). The result of this section is a co-located data sets of 2704 samples, which consists of

MERRA-2 ALHs and OMAERUV observations that are quality-assured by the AERONET records.

### 4.3  UVAI dependence on MERRA-2 ALHs

Fig.7 presents the MERRA-2 ALHs against the OMAERUV UVAI as a function of the MERRA-2 AOD. For each ALH

parameter, we categorize the data into four groups by AOD values that are indicated by the title of plots in the first row. The

magnitude of AOD is also indicated by the marker size. For each AOD group, we provide the number of samples (N), the

spearman correlation coefficient (R) and the slope of linear regression (k) between ALH and UVAI. The latter two are used

to quantify the UVAI altitude dependence.

The performances of the three effective heights ($H_{aer}^{\beta}$, $H_{aer}^{\tau}$ and $H_{aer}^{63}$) are similar to each other. They all show relatively

strong but negative correlation (R is around -0.4) with UVAI in the first AOD group (AOD smaller than 0.2). As discussed

in Section 2.4, the effective heights show more variabilities at low AOD situations. The negative correlation may be caused

by that the effective heights tend to get extremely high values for very low aerosol loading cases, for instance, the aerosol

profiles in Antarctic as explained in Fig.4c. The correlation coefficients and the slope between the UVAI and the three

effective heights increase with aerosol loading in the next two AOD regimes, however, barely no correlation is found under

the last AOD group. Although the UVAI varies from 0 to 5, the effective heights are less variable in the highest AOD

regime. It is because that the peak extinctions are most likely to appear in the lower part of the atmosphere (below 3 km,

Fig.3), resulting in the derived ALHs are limited below this altitude.

Compared with effective counterparts, the overall correlation between UVAI and $H_{aer}^{t}$ is higher. Its correlation with UVAI

increases with aerosol loading and so does the linear fitting slope, which is consistent with our knowledge that UVAI

dependence on ALH increases with AOD as explained in Section 4.1. But whether this UVAI dependence on ALH is

realistic needs further validation with observations, which will be discussed in the following sections.



## 4.4 UVAI dependence on satellite ALH products

Fig.8-11 present the relationship between the selected satellite ALH products and their corresponding UVAI. Similar to Fig.7, we group the data into four regimes by the AOD values. Note that the AOD thresholds for each group differ from one product to another.

Fig.8 presents the OMAERUV ALH from the AERONET quality-screened OMAERUV-MERRA-2 joint data set (as described in Section 4.2). The OMAERUV ALH is overall positively correlated with UVAI, and both the correlation and the slope increase with AOD. It is noted that the correlation between OMAERUV ALH and UVAI is strong (R is larger than 0.6). As described in Section 3.1, the OMAERUV ALH is specifically designed to detect the height of absorbing aerosol layers, and the UVAI information also involves in the determination of ALH (Torres et al., 2013). Furthermore, the quality

control by the AERONET records may further enhance the correlation between UVAI and ALH. Consequently, the correlation may be overestimated compared to the reality.

Fig.9 shows the relationship between the OMI $O_2$-$O_2$ ALH trained for SSA of 0.90 and 0.95 and the corresponding OMAERUV UVAI. The AOD here is accompanied with the OMI $O_2$-$O_2$ ALH products, which is the co-located MODIS AOD used in NN prediction. Although we only use samples with AOD larger than 1 in order to exclude potential outliers in

the retrieved ALH due to the low $O_2$-$O_2$ absorption signal, there are still some extreme values (above 10 km), especially for the lowest AOD regime. The likely reasons could be the mismatch of aerosol models between the training and prediction process, and/or the clouds contamination in sub-pixels. The magnitude of the correlation and slope between the $O_2$-$O_2$ ALH and UVAI are negligible until the last AOD group. Apart from the outliers, this may be caused by that the product excludes majority of dust aerosols over land (one of the most important absorbing aerosol sources) due to the bright surface.

Moreover, unlike other ALH products that more or less use UVAI as an entry in retrieval algorithms, the $O_2$-$O_2$ ALH retrieval is purely independent to information of UVAI.

Since currently there is no operational AOD product for TROPOMI, we use the Dark Target and Deep Blue combined AOD at 550 nm provided by the MODIS Level 3 Daily Atmosphere Gridded Product (King et al., 2013). The AOD collected from the Aqua platform (overpass time: 13:30 local time, MYD08_D3, http://dx.doi.org/10.5067/MODIS/MYD08_M3.006,

last access: 15 December 2019) is co-located to the TROPOMI ALH. The relationship between the TROPOMI ALH and the accompanied level 2 TROPOMI UVAI level 2 product is shown in Fig.10. Note that the TROPOMI ALH is only retrieved for pixels with UVAI (calculated by 354-388 nm wavelength pair) larger than 1. The overall UVAI altitude dependence is lower than that of OMAERUV. Nevertheless, both the correlation and the slope increase with AOD from 0.05 to 0.39 and 0.08 to 0.96, respectively.

Similar to TROPOMI, as there is no operational AOD product of GOME-2, we co-locate the Dark Target and Deep Blue combined AOD at 550 nm provided by the MODIS level 3 Daily Atmosphere Gridded Product collected from the Terra platform (overpass time: 10:30 local time, MOD08_D3, http://dx.doi.org/10.5067/MODIS/MOD08_M3.006, last access: 15 December 2019) to the GOME-2 AAH. The GOME-2 AAH retrieves ALH for pixels with the accompanied level 2





GOME-2 UVAI (calculated by 340-380 nm wavelength pair) larger than 2 though, it is suggested to use pixels with UVAI
larger than 4 to assure reliable retrievals. Besides, only samples with could fraction lower than 0.25 are retained due to
higher reliability (Tilstra et al., 2018). Fig.11 presents the GOME-2 AAH against the corresponding UVAI. The UVAI
dependence on ALH is positively associated with AOD with the increasing correlation from 0.17 to 0.49 and the increasing
slope from 0.10 to 0.38.

### 4.5  Comparison between the MERRA-2 ALHs and the satellite ALHs

In this section, we only focus on the correlation coefficient between UVAI and different ALH data sets, as the magnitude of
the slope depends on each data set. The correlation between UVAI and above ALH data sets from the model as well as
measurements are summarized in Fig.12. As the AOD thresholds varies with data sets, we assign each AOD regime with a
group number. The AOD group 1 indicates the lowest AOD regime and the AOD group 4 indicates the highest one.
According to Fig.12, UVAI is positively associated with the OMAERUV ALH, the TROPOMI ALH and the GOME-2
AAH. Reflected by the increasing correlation, the UVAI altitude dependence on these products becomes stronger with
increasing AOD. On the other hand, UVAI is weakly correlated with the OMI $O_2$-$O_2$ ALH except for the highest AOD
regime. The reason could be that the $O_2$-$O_2$ ALH algorithm is sensitive to outliers associated with relatively low aerosol
loadings. Furthermore, the data availability of the one-year OMI $O_2$-$O_2$ product is much less than others (Table 2), and
excludes dust aerosols, a major absorbing aerosol type. Besides, the $O_2$-$O_2$ ALH is totally independent to UVAI information
during the retrieval, while the rest ALH products are more or less dependent on the UVAI as introduced in Section 3.
For the majority ALH products, the coefficients between UVAI and ALH are close to 0 at the lowest AOD regimes and
eventually approaches around a level of about 0.4. The OMAERUV ALH is an exception, because the UVAI is intentionally
input to the ALH retrieval, and the data is quality assured with the AERONET records. Thus, it shows the strongest
correlation with UVAI for all AOD regimes (R is larger than 0.6), which may not be realistic.
The UVAI correlation with the TROPOMI ALH is similar to that with the GOME-2 AAH. The reason could be that both
ALH retrievals are based on the $O_2$ absorption properties over the near-infrared band, and they both use the clouds
information retrieved by FRESCO algorithm, and they both are obtained during a similar period. Their correlation with the
corresponding UVAI increases with AOD, which is consistent with our a-priori knowledge on the altitude dependence of
UVAI in Section 4.1. Although the correlation between UVAI and these two ALH data sets may be slightly overestimated as
a certain UVAI threshold is applied (1 for TROPOMI and 4 for GOME-2) to filter out low aerosol loadings or weak-
absorbing scenes, the correlation coefficient ranges from near 0 to around 0.4. This value may be more realistic than that of
the high correlation between the OMAERUV ALH and UVAI (R is larger than 0.6 for all AOD regimes).
Among all the MERRA-2 ALHs, only the relationship between the top boundary height ($H_{aer}^t$) and UVAI matches that the
altitude dependence of UVAI increases with AOD. The correlation coefficient between $H_{aer}^t$ and UVAI increases from 0.14
to 0.42, which is in good agreement with that between UVAI and TROPOMI ALH / GOME-2 AAH. On the other hand, the





effective heights ($H_{aer}^{\beta}$, $H_{aer}^{\tau}$ and $H_{aer}^{63}$) show high variabilities and sometimes predict extremely high ALH for low AOD cases, while they are less variable when the aerosol loading is high. Consequently, all these three parameters show strong negative correlation between UVAI and ALH in the lowest AOD regimes, while little correlation for high aerosol loading conditions.

In summary, among the selected ALH products, the TROPOMI ALH and the GOME-2 AAH are the two preferred candidate products for UVAI analysis. Their correlation with UVAI is generally positive and increases with aerosol loading, which agrees with our theoretical knowledge on UVAI as described in Section 4.1. By comparing the MERRA-2 ALHs with the satellite products, we find that the relationship between $H_{aer}^{t}$ and the corresponding UVAI is most similar to the two satellite ALH products. In other words, the MERRA-2 $H_{aer}^{t}$ can be an alternative data set to interpret aerosol absorption from UVAI

in case that there is no observational aerosol vertical distribution available.

## 5    Conclusions

Aerosol vertical distributions are important to aerosol radiative forcing assessments and atmospheric remote sensing research. From our perspective, ALH is necessary to quantitively analyze aerosol absorption from UVAI. Facing the problem that observations of aerosol vertical distributions are still limited, this paper attempts to find an ALH data set for

UVAI analysis using aerosol extinction profiles provided by the MERRA-2 aerosol reanalysis.

We proposed four methods to derive ALH from given aerosol extinction profiles: (1) the extinction-weighted mean height ($H_{aer}^{\beta}$), (2) the AOD-weighted mean height ($H_{aer}^{\tau}$), (3) the scale height ($H_{aer}^{63}$) and (4) the top boundary height ($H_{aer}^{t}$). We tested their sensitivity to the extinction profiles and found that the $H_{aer}^{t}$ is the most robust. The test also provided the following implications which may be useful for deriving ALH from aerosol extinction profiles in future studies:

(1) Whether an aerosol extinction profile is evenly gridded in vertical direction has a significant impact on $H_{aer}^{\beta}$;

(2) A coarse vertical grid resolution (larger than 0.5 km) may bias all the ALH quantities;

(3) If an aerosol extinction profile only covers troposphere, the derived ALH may differ from that derived from a full profile. If there are no major volcanic eruptions, aerosol loadings above 20 km is negligible;

(4) The minimum extinction coefficient (detection limit of instruments or background values of models) matters when it is

over 0.001 km⁻¹.

The spatial-temporal patterns of MERRA-2 $H_{aer}^{t}$ is generally in agreement with variations in AOD. The major aerosol sources, however, are not reflected by the distribution of $H_{aer}^{\beta}$, $H_{aer}^{\tau}$ and $H_{aer}^{63}$. The three effective ALH quantities tend to predict a high altitude where little aerosol exists, or an altitude close to the surface under high aerosol loading. The reason is that their calculations are sensitive to the extinction lapse rate, the magnitude and location of peak extinction layers.

In the second part of this paper, we examined the MERRA-2 ALHs and four satellite ALH products in terms of their relationship with UVAI. The four ALH products are correlated with the corresponding UVAI to some extents. The OMI O₂-



O$_2$ ALH has a relatively weaker correlation with UVAI than others, especially when AOD is low, since it is sensitive to outliers at low O$_2$ absorption signals. Besides, the OMI O$_2$-O$_2$ ALH excludes most of dust aerosols over land and its retrieval does not depends on UVAI information. By contrast, the ALH reported in the official OMAERUV product shows the highest
correlation with UVAI which may be considered to be overestimated. This is because the OMAERUV ALH is intentionally designed for absorbing aerosols and uses UVAI as an input in its retrieval. The quality assurance procedure by the AERONET records also enhances the correlation. The TROPOMI ALH and the GOME-2 AAH behaves similarly as they are retrieved in the same band and use the same cloud inputs. These two considered as preferred candidate ALH products, since they both show a gradually stronger correlation with UVAI as AOD increases, which agrees with our a-priori
knowledge of UVAI altitude dependence. The correlation coefficients are moderate (R is around 0.4-0.5) in the highest AOD regime, which is more reasonable compared with the strong correlation between the OMAERUV UVAI and ALH. Among all MERRA-2 ALH parameters, only the aerosol layer top boundary height ($H_{aer}^t$) matches the behavior of the TROPOMI ALH and the GOME-2 AAH.

In summary, the definition of the MERRA-2 aerosol layer top boundary height ($H_{aer}^t$) proposed in this paper is robust to
changes in the given aerosol extinction profiles. In other words, this method can be applied to aerosol profiles provided by other data sets. The spatial distribution and temporal variation of $H_{aer}^t$ derived from the MERRA-2 aerosol fields are well associated aerosol sources and atmospheric dynamics. More importantly, UVAI dependence on this quantity increases with AOD, and the magnitude of their correlation coefficient matches that we found from the observational data sets, i.e. the TROPOMI ALH and the GOME-2 AAH. This means in case that there is no ALH information provided by TROPOMI or
GOME-2, $H_{aer}^t$ derived from MERRA-2 can be an alternative ALH data source and contribute to interpret the aerosol absorption from UVAI observations.

**Appendix A: converting from aerosol mass concentration profiles to aerosol extinction coefficient profiles**

The MERRA-2 3-hourly instantaneous aerosol mass mixing ratio (MERRA-2 inst3_3d_aer_Nv, 10.5067/LTVB4GPCOTK2 , last access: 7 June 2019) provides mass mixing ratio profiles for 15 aerosol sub-species. The
conversion from mass concentrations to extinction coefficients is as follows:

$$\rho_x(z) = c_x(z) \times \rho_{air}(z) \tag{A1}$$

$$\beta_x(z) = \rho_x(z) \times \beta_{m,x}(z) \tag{A2}$$

, where $c_x(z)$ is the mass mixing ratio (unit: kg kg$^{-1}$) of an aerosol type $x$ at altitude $z$ (unit: m). $\rho_{air}(z)$ and $\rho_x(z)$ are the mass density (unit: kg m$^{-3}$) of air and the aerosol species $x$. $\beta_{m,x}(z)$ and $\beta_x(z)$ are the mass extinction coefficients (unit: m$^2$ kg$^{-1}$) and extinction coefficients (unit: m$^{-1}$) for the aerosol species $x$. The aerosol mass extinction coefficient $\beta_{m,x}(z)$ is a function of relative humidity, whose values are provided in the supplementary document of Randles et al. (2017). The total
extinction profile $\beta(z)$ is the summation of $X$ aerosol species.



$$\beta(z) = \sum_{x=1}^{X} \beta_x(z) \qquad \text{(A3)}$$

The altitude (MERRA-2 inst3_3d_asm_Np 10.5067/WWQSXQ8IVFW8 , last access: 7 June 2019). The terrain height ($z_0$, unit: m) is converted from temporally constant surface geo-potential ($\phi_0$, unit: m²s⁻²) (MERRA-2 const_2d_ctm_Nx, 10.5067/4Z3YUPM81GRJ, last access: 4 March 2019).

$$z_0 = \frac{\phi_0}{g_0} \qquad \text{(A4)}$$

, where $g_0$ is the standard gravity (unit: ms⁻²) at mean sea level, which is about 9.81 ms⁻².

**Appendix B: validating MERRA-2 aerosol reanalysis with CALIOP measurements**

To prove that the MERRA-2 aerosol fields are realistic, we validate them with CALIOP measurements. CALIOP is designed to measure vertical profiles of elastic backscatter at two wavelengths (1064 nm and 532 nm) from a near nadir-viewing geometry during both daytime and nighttime. Accurate aerosol and cloud heights and the high-resolution (333 m and 30 m in horizontal and vertical direction) extinction coefficient profiles are derived from the total backscatter measurements (Winker

et al., 2006). We employ the CALIOP level 3 all-sky aerosol extinction profiles from 2006 to 2016 (http://10.5067/CALIOP/CALIPSO/CAL_LID_L3_APro_AllSky-Standard-V3-00 , available from 2006-06 to 2016-11, and http://10.0.19.203/CALIOP/CALIPSO/CAL_LID_L3_APro_AllSky-Standard-V3-10 , available from 2016-12 onwards, last access: 15 April 2019) to validate the MERRA-2 extinction profiles in the troposphere (below 12 km). The level 3 climatology is monthly available from June 2006 onwards. The vertical and horizontal resolution is 60 m and 2° ×

5° (latitude by longitude), respectively.

Fig.B1 shows the seasonal zonal aerosol extinction coefficient profiles ($\beta(z)$) as a function of latitude and Fig.B2 shows the seasonal AOD maps calculated from the extinction coefficients below 12 km. The magnitude of the MERRA-2 extinction coefficients in the free troposphere and AOD are generally higher than that of CALIOP. It may because that the MERRA-2 aerosol fields seem to have higher background level of extinction coefficients. Besides, the CALIOP measurements suffer

from problems of missing data or attenuated signal due to presence of clouds or over bright surfaces, etc. The largest differences occur at the Sahara region and biomass burning region in south Africa during fall, and smoke plume over the southern Atlantic Ocean during summer and fall. Nevertheless, the spatial distribution and temporal variation of MERRA-2 aerosol fields generally agree well with that of CALIOP, indicating the MERRA-2 aerosol assimilation system works properly.

Both MERRA-2 and CALIOP data show that, in most situations, aerosols are located near the ground (below 3 km). Aerosols mainly come from low/mid-latitudes continents in the Northern Hemisphere, i.e. China, India, and Saudi Arabian Peninsula and central and North Africa. The major sources in the Southern Hemisphere are the biomass burning regions in south Africa during summer and south America during autumn.



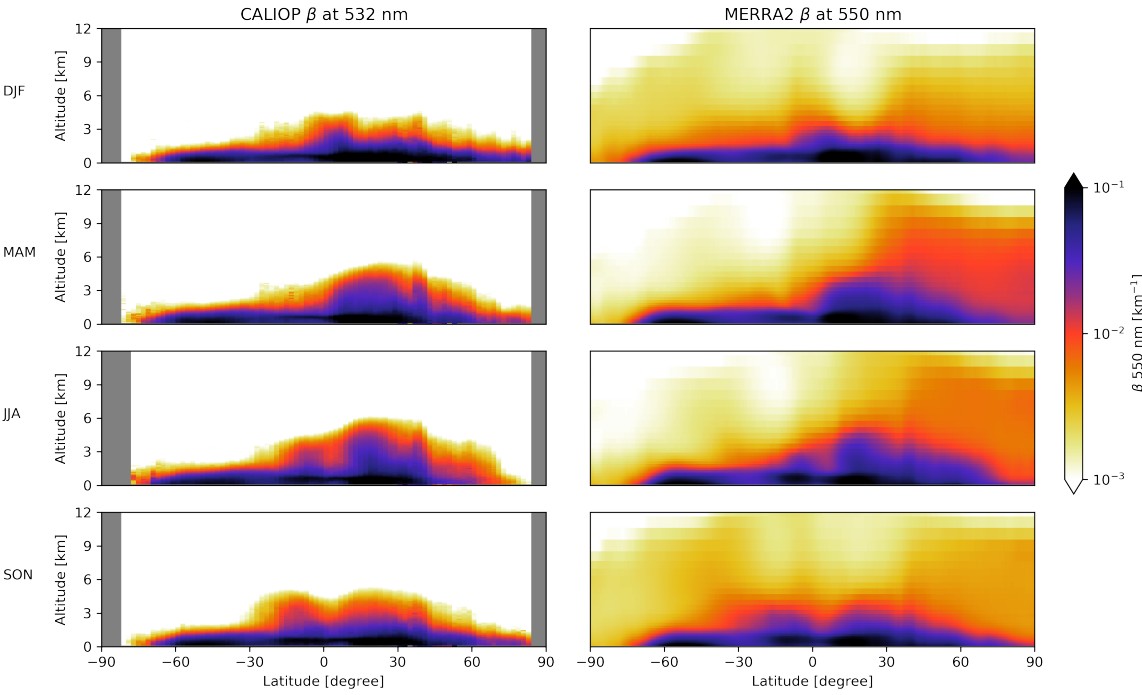

**Figure. B1 Seasonal extinction coefficient profile as a function of latitude of CALIOP (left column) and MERRA-2 (right column) during period from 2006 to 2016.**



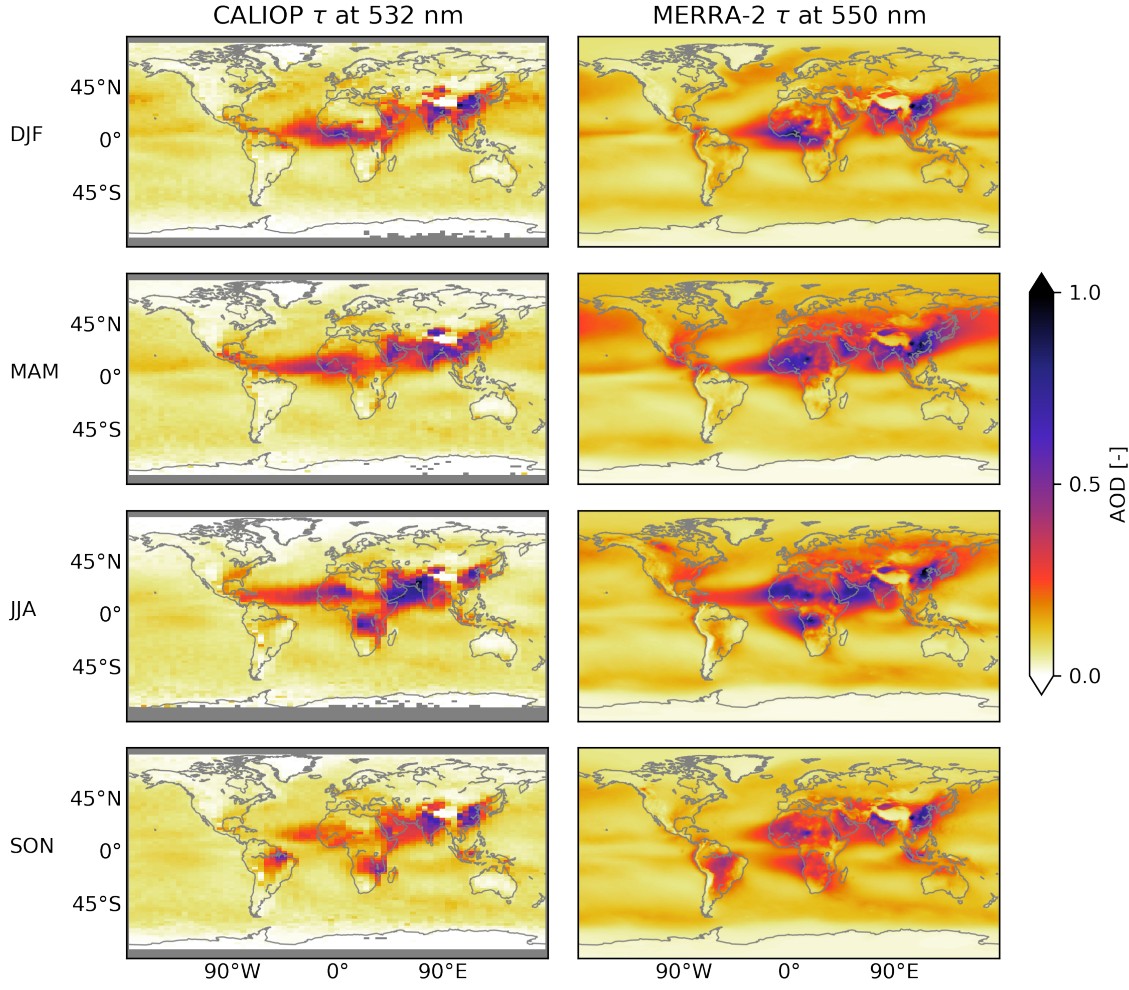

**Figure. B2 Seasonal AOD climatology of CALIOP (left column) and MERRA-2 (right column) during period from 2006 to 2016.**

**Appendix C: the choice of extinction lapse rate threshold to determine the aerosol layer top height**

The selected threshold of the extinction lapse rate ($\gamma_{ext}$) determines the top boundary height of an aerosol layer ($H_{aer}^t$). According to our definition, above this altitude, $\gamma_{ext}$ should always no more than the pre-selected threshold. The choice of the threshold is empirical, which is based on the following sensitivity study.

As shown in Fig.C1, we select four regions of interest and use their mean extinction profiles ($\beta(z)$) as representative profile shapes, i.e. East China, North Africa, Antarctica and South Africa. For each region, the mean profile shape (black lines) is obtained from the average of extinction profiles during the period from 2006-01-01 to 2016-12-31. We also plot the standard deviation to show the variation of the extinction profiles (gray bars), and the extinction lapse rate to show how fast the





extinction profiles changes with height (blue lines). Next, we calculate $H_{aer}^{t}$ using different $\gamma_{ext}$ ranging from 0.001 to 0.1

km$^{-2}$. According to Fig.C1, it is clear that a threshold between 0.005 and 0.01 km$^{-2}$ may be suitable for all profile shapes.

Thus, in this paper, we use the threshold value of 0.01 km$^{-2}$.

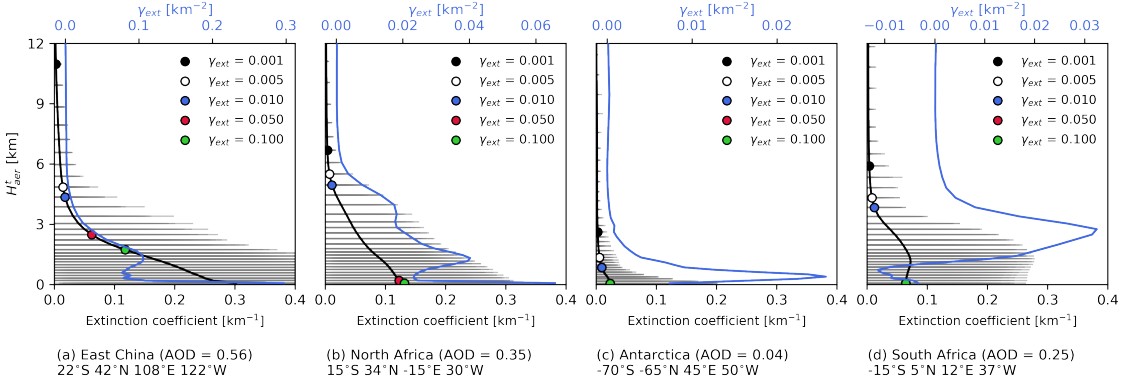

**Figure. C1 Sensitivity study to determine the threshold of the extinction lapse rate ($\gamma_{ext}$) for the aerosol layer top boundary height ($H_{aer}^{t}$). The sensitivity study is conducted for four representative aerosol extinction coefficient profiles ($\beta(z)$) for (a) East China,**

**(b) North Africa, (c) Antarctica and (d) South Africa. The black lines are the mean profiles and gray bars are the standard deviation during the period from 2006-01-01 to 2016-12-31. The blue lines are the $\gamma_{ext}$ of the mean profile.**

### Appendix D: Configurations of radiative transfer simulations of UVAI

This section presents the configurations of a sensitivity study showing how UVAI response to ALH under different situations. The UVAI is simulated by the radiative transfer model Determining Instrument Specifications and Analysing

Methods for Atmospheric Retrieval (DISAMAR) developed by KNMI (de Haan, 2011). The aerosols are described by the Henyey-Greenstein phase function with constant asymmetry parameter of 0.7. The wavelength dependence of AOD is characterized by the Ångström exponent of 1. The single scattering albedo is chosen between 0.9 (relatively absorbing aerosols) and 0.99 (scattering aerosols). The AOD varies from 0.01 to 3. The aerosol layer is parameterized as a box-shape profile with a constant geometric depth of 50 hPa and a varying middle altitude from 0.5 to 12 km. The surface albedo is

specified at 388 nm but also set to be spectrally flat. Other inputs are listed in Table D1.

**Table D1 Configurations of UVAI radiative transfer simulations.**

| Parameter | Value |
| --- | --- |
| Aerosol optical thickness at 550 nm (AOD$_{550}$) | From 0.01 to 3 |
| Aerosol layer pressure [hPa] | From 200 to 950 hPa (approximately 0.5 to 12 km) |
| Aerosol layer thickness [hPa] | 50 |
| Single scattering albedo at 550 nm (SSA$_{550}$) | 0.90, 0.99 |





| Asymmetry factor | 0.7 |
|---|---|
| Ångström exponent coefficient | 1 |
| Solar zenith angle (SZA) [°] | 30 |
| Viewing zenith angle (VZA) [°] | 0 |
| Relative azimuth angle (RAA) [°] | 120 |
| Surface pressure [hPa] | 1013 |
| Surface albedo at 388 nm ($As_{388}$) | 0.03, 0.1 |

**Acknowledgement**

This work was performed in the framework of the KNMI Multi-Annual Strategic Research (MSO). The authors thank NASA's GES-DISC for free online access to the MERRA-2 aerosol reanalysis and the OMAERUV data, thank LAADS DAAC for free online access to the MODIS data, thank ESA's Copernicus Open Assess Hub for free online access to the TROPOMI data, and thank NASA Goddard Space Flight Center AERONET project for providing the data from the AERONET.

**Author contributions**

JS provided the scientific idea, processed the data and led the writing of the paper; LGT, JC and SN were responsible to provide the data and check the section of the GOME-2 absorbing aerosol layer height, the OMI $O_2$-$O_2$ neural network aerosol layer height and the TROPOMI $O_2$ A-band aerosol layer height, respectively. JPV, PvV and PL provided the guidance in writing the paper.

**Competing interests**

The authors declare that they have no conflict of interest.

**Data availability**

The MERRA-2 3-hourly instantaneous aerosol mass mixing ratio (MERRA-2 inst3_3d_aer_Nv) can be accessed via 10.5067/LTVB4GPCOTK2 (last access: 7 June 2019), the corresponding MERRA-2 altitude information (MERRA-2
inst3_3d_asm_Np ) can be accessed via 10.5067/WWQSXQ8IVFW8  (last access: 7 June 2019) and the terrain height (MERRA-2 const_2d_ctm_Nx) can be accessed via 10.5067/4Z3YUPM81GRJ (last access: 4 March 2019). The CALIOP level 3 all-sky aerosol extinction profiles can be accessed via



http://10.5067/CALIOP/CALIPSO/CAL_LID_L3_APro_AllSky-Standard-V3-00 and

http://10.0.19.203/CALIOP/CALIPSO/CAL_LID_L3_APro_AllSky-Standard-V3-10 (last access: 15 April 2019). The

OMI/Aura level 2 OMAERUV product can be accessed via http://dx.doi.org/10.5067/Aura/OMI/DATA2004 (last access: 12

August 2019). The TROPOMI aerosol layer height level 2 offline data can be accessed via

https://s5phub.copernicus.eu/dhus/#/home (last access: 18 November 2019). The AERONET version 3 level 1.5 direct-sun

and inversion products can be accessed via https://aeronet.gsfc.nasa.gov (last access: 18 November 2019). The MODIS daily

level 3 AOD product from the Aqua (MYD08_D3) and the Terra (MOD08_D3) platform can be accessed via

http://dx.doi.org/10.5067/MODIS/MYD08_M3.006 and http://dx.doi.org/10.5067/MODIS/MOD08_M3.006,

respectively (last access: 15 December 2019). The MERRA-2 aerosol layer heights derived in this paper, the OMI $O_2$-$O_2$

neural network aerosol layer height (last access: 15 July 2019) and the GOME-2 absorbing aerosol layer height (last access:

24 November 2019) are available from authors upon requests. If you are interests, please send message to

jiyunting.sun@knmi.nl for the MERRA-2 aerosol layer height data sets, to gijsbert.tilstra@knmi.nl for the GOME-2

absorbing aerosol layer height and to Julien.Chimot@eumetsat.int for the OMI $O_2$-$O_2$ neural network aerosol layer height.

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

**Table 1 Definitions of aerosol layer height in this paper.**

| Aerosol layer height | Symbols | Derivation method |
|---|---|---|
| Extinction-weighted mean aerosol layer height | $H_{aer}^{\beta}$ | Eq.(1) |
| AOD-weighted mean aerosol layer height | $H_{aer}^{\tau}$ | Eq.(2) |
| Aerosol scale height with 63% AOD present | $H_{aer}^{63}$ | Eq.(3) |
| Aerosol layer top height | $H_{aer}^{t}$ | The first height searched from the surface where $|\gamma_{ext}(z)|<0.01$ for $z \geq H_{aer}^{t}$ |

**Table 2 Information on the selected satellite ALH products in this paper.**

| Data product | Short name | Time period | Number of observations after pre-processing |
|---|---|---|---|
| ALH reported in OMAERUV | OMAERUV ALH | 2006-01-01 to 2016-12-31 | 105,392,332 |
| OMI $O_2$-$O_2$ neural network ALH | $O_2$-$O_2$ ALH | 2006-01-01 to 2006-12-31 | 58,172 (SSA = 0.90), 67,718 (SSA = 0.95) |
| TROPOMI $O_2$ A-band ALH | TROPOMI ALH | 2018-11-13 to 2019-08-31 | 430,414 |
| GOME-2 absorbing aerosol layer height | GOME-2 AAH | 2018-08-13 to 2019-10-31 | 431,176 |




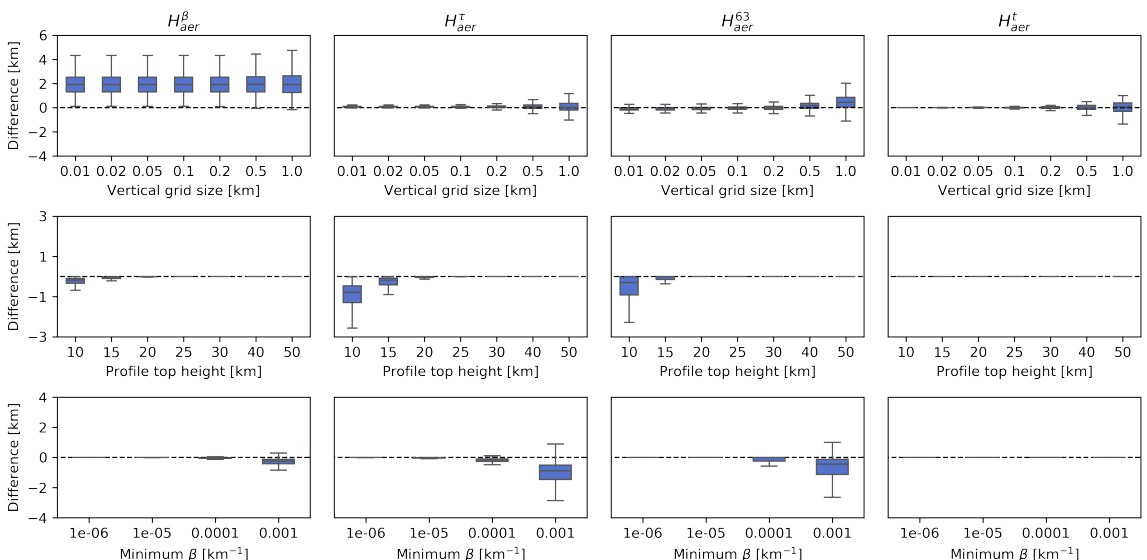

**Figure.1 Sensitivity of different ALH definitions to the properties of a given aerosol extinction profile. Total 401,800 profiles are selected. Each column indicates different ALH definitions, and each row indicates ALH sensitivity to how an extinction profile is gridded, to which altitude an extinction profile extends, and what is the minimum extinction coefficient. The boxplot is the statistics of difference between the ALH derived from original profiles and that derived from modified ones (the latter minus the former).**

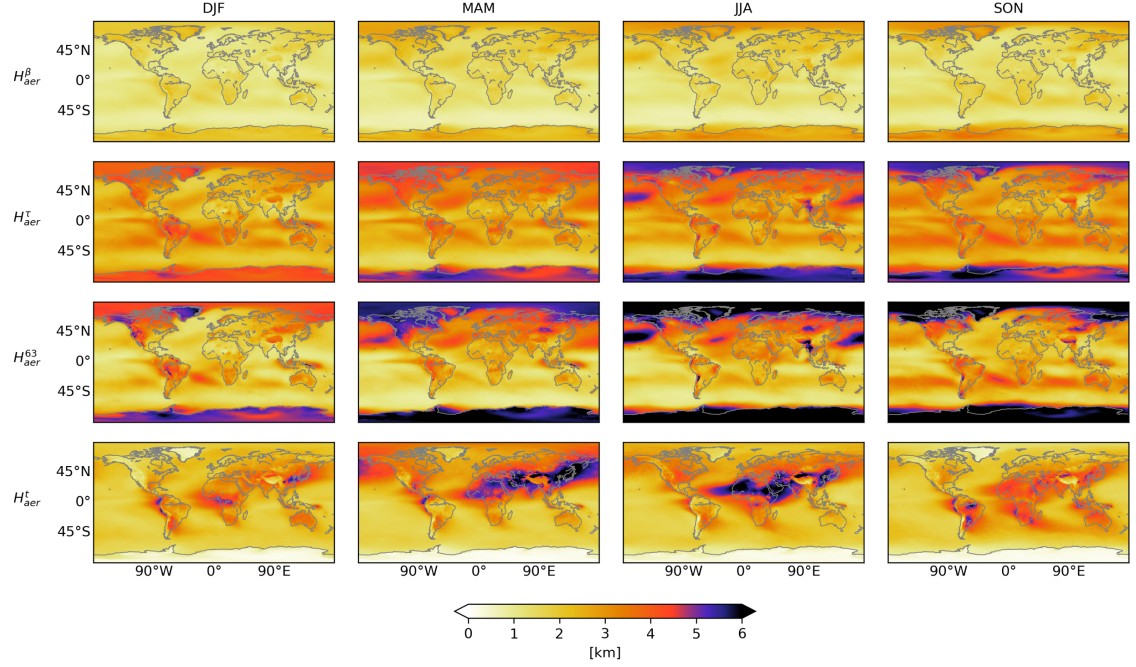





**Figure.2 Spatial distribution maps of the MERRA-2 ALHs seasonal climatology during period from 2006-01-01 to 2016-12-31. Rows represent different ALHs definition and columns represent different seasons.**


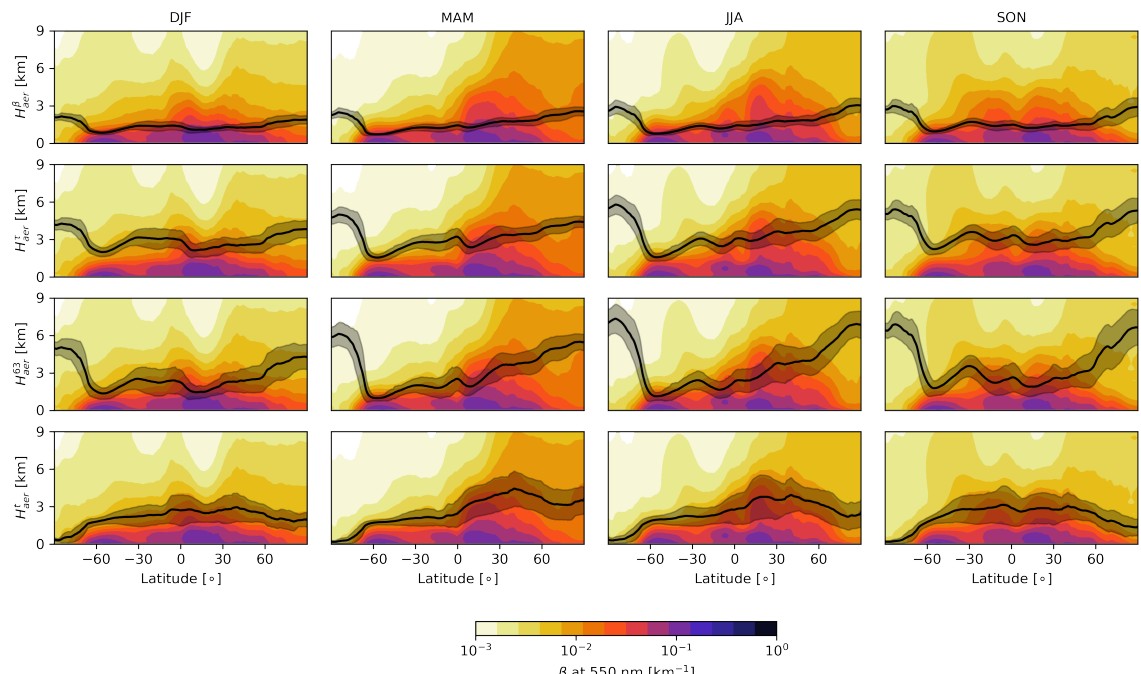

**Figure.3 The zonal average (black solid line) and the standard deviation (grey filling area) of the MERRA-2 ALHs and the zonal average of the MERRA-2 extinction coefficients (contours) for the period from 2006-01-01 to 2016-12-31. Rows represent different ALHs definition and columns represent different seasons.**


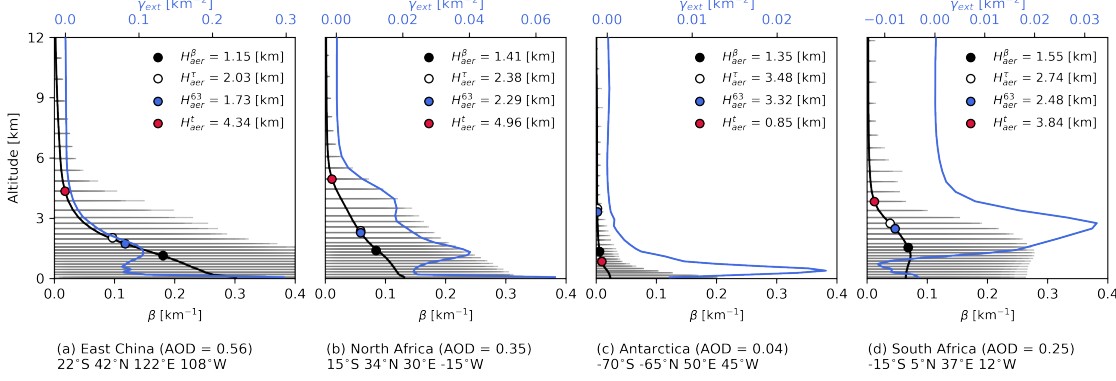

**Figure.4 Representative aerosol extinction coefficient profiles for (a) East China, (b) North Africa, (c) Antarctica and (d) South Africa. The black lines are the mean profiles and gray bars are the standard deviation during the period from 2006-01-01 to 2016-12-31. The blue lines are the extinction lapse rate ($\gamma_{ext}$) of the mean profile.**






**Figure.5** Data availability of four satellite ALH products used in this paper. Satellite data is gridded onto the MERRA-2 coordinates of 0.5° × 0.625°. Left column: the mean of each ALH product during the corresponding period; middle column: the corresponding mean UVAI; right column: number of observations during the corresponding period. The color bar is normalized 870 to unit by dividing with maximum values indicated in each plot.




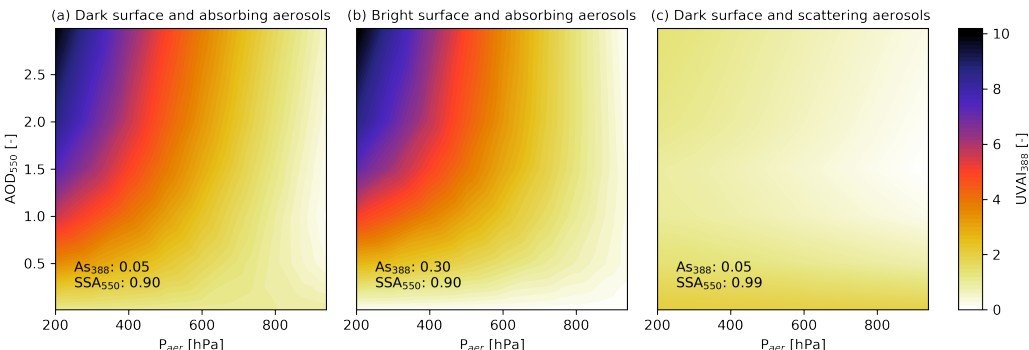

**Figure.6 UVAI dependence on ALH under (a) a dark surface and absorbing aerosols; (b) a bright surface and absorbing aerosols; (c) a dark surface and scattering aerosols.**


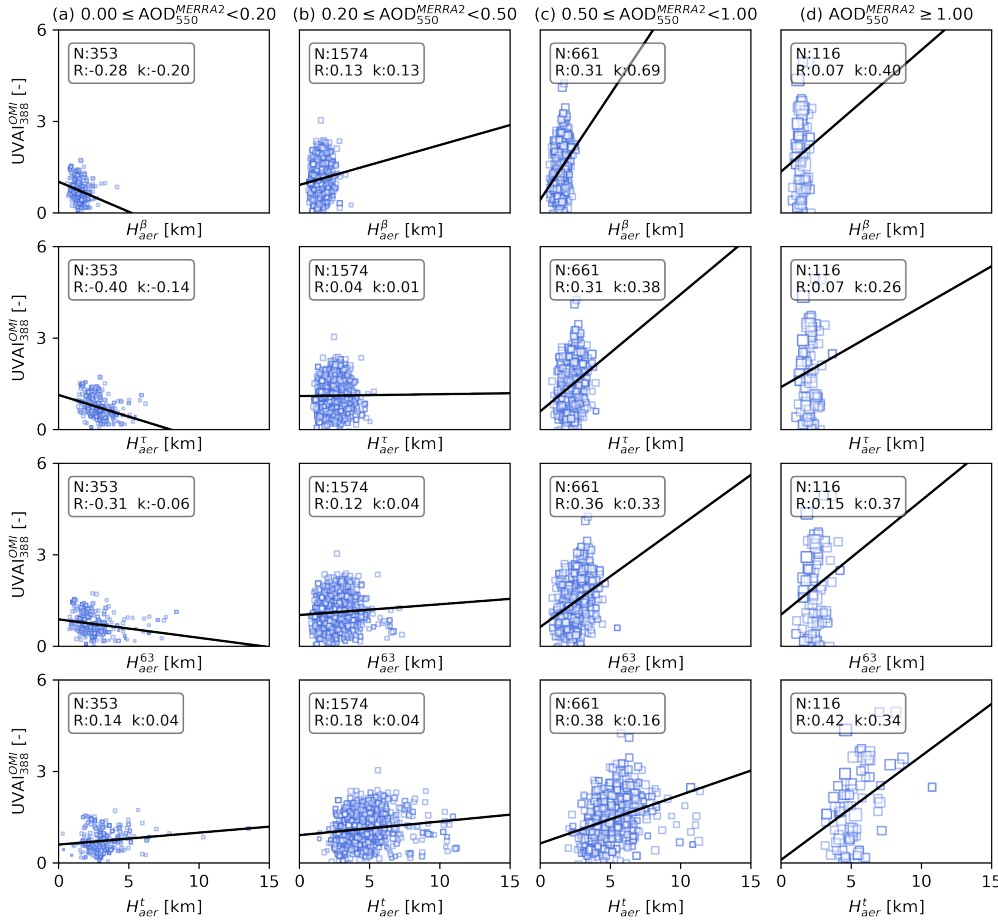




**Figure.7 The MERRA-2 ALHs against the OMAERUV UVAI as a function of the MERRA-2 AOD. The marker size indicates the magnitude of AOD. For each AOD regime, the number of samples (N), the spearman correlation coefficient (R) and the slope of linear regression (k) between ALH and UVAI are provided.**


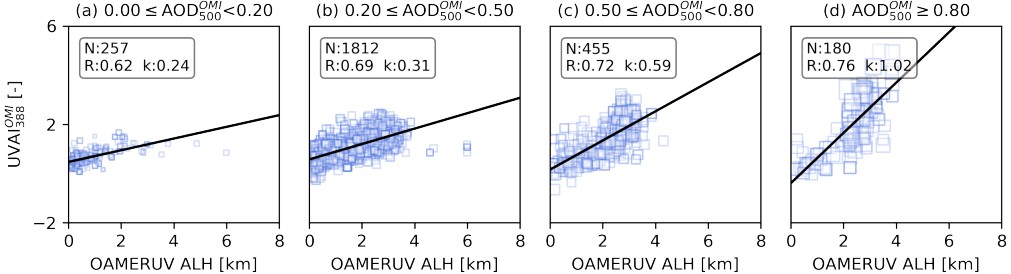

**Figure.8 The OMAERUV ALH against the OMAERUV UVAI as a function of the OMAERUV AOD. The marker size indicates the magnitude of AOD. For each AOD regime, the number of samples (N), the spearman correlation coefficient (R) and the slope of linear regression (k) between ALH and UVAI are provided.**


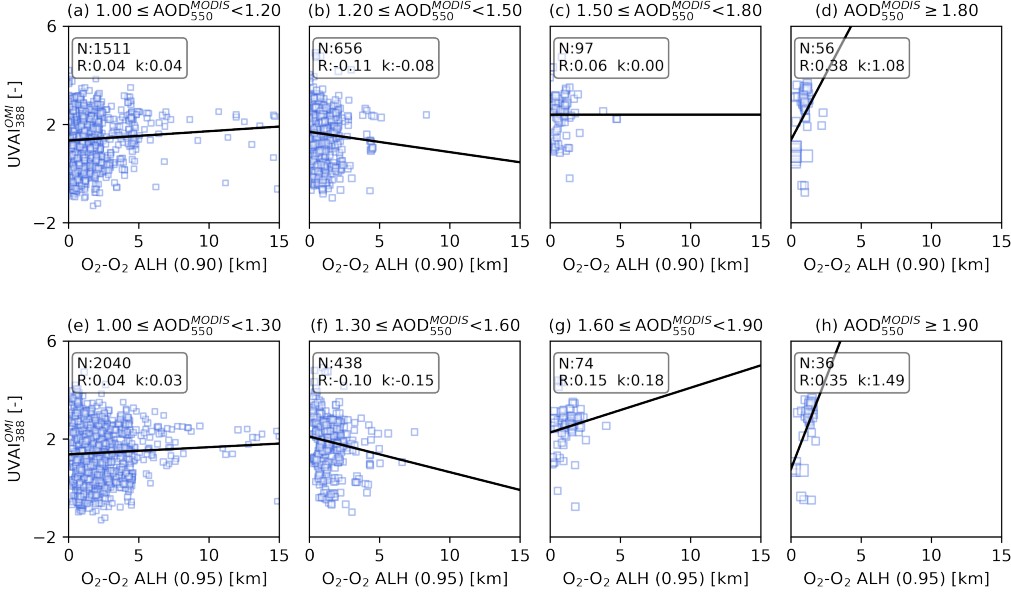

**Figure.9 The OMI O$_2$-O$_2$ ALH against the OMAERUV UVAI as a function of the MODIS AOD for SSA = 0.90 (first row) and SSA = 0.95 (second row). The marker size indicates the magnitude of AOD. For each AOD regime, the number of samples (N), the**

**spearman correlation coefficient (R) and the slope of linear regression (k) between ALH and UVAI are provided.**



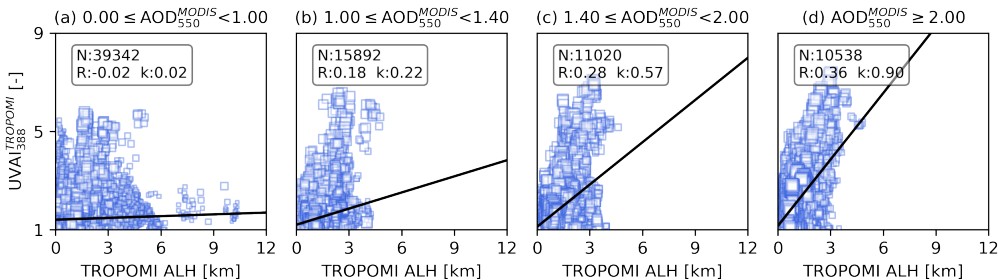

**Figure.10 The TROPOMI O₂ A-band ALH against the TROPOMI UVAI as a function of the MODIS AOD. The marker size indicates the magnitude of AOD. For each AOD regime, the number of samples (N), the spearman correlation coefficient (R) and the slope of linear regression (k) between ALH and UVAI are provided.**

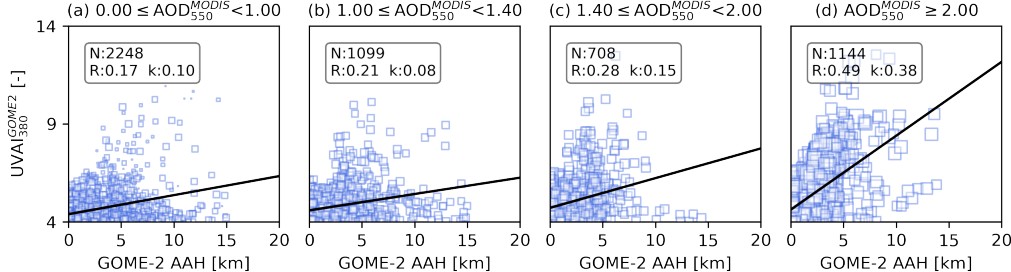

**Figure.11 The GOME-2 AAH against the GOME-2 UVAI as a function of the MODIS AOD. The marker size indicates the magnitude of AOD. For each AOD regime, the number of samples (N), the spearman correlation coefficient (R) and the slope of linear regression (k) between ALH and UVAI are provided.**

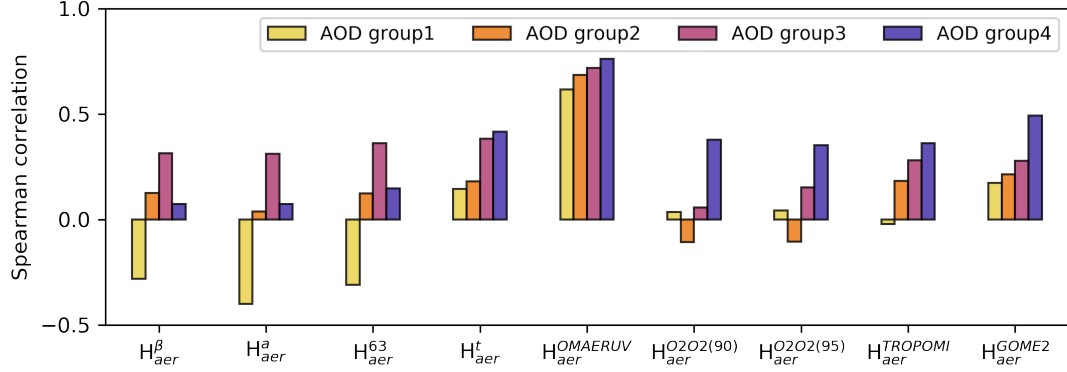

**Figure.12 Spearman correlation coefficients between ALH and UVAI as a function of AOD. The magnitude of AOD is monotonically increasing from group 1 to group 4.**