# Peer review of "Defining aerosol layer height for UVAI interpretation using aerosol vertical distributions characterized by MERRA-2"

_Atmospheric Chemistry and Physics, 2020_

## Referee Comment (RC1) · Anonymous Referee #2 · 23 Mar 2020

**Review of ACP-2020-39 article '*Defining aerosol layer height for UVAI interpretation using aerosol vertical distributions characterized by MERRA-2*', by Sun et al.**

**Summary**

The work described in this manuscript proposes the use of the multi-year long MERRA-2 data base on aerosol vertical distribution from model calculations to derive a long-term record of Aerosol Layer Height (ALH) that can be used in the interpretation of the multi-decadal UV Aerosol Index (UVAI) from a series of instruments, from TOMS in 1979 to currently operational OMI, GOME and TROPOMI sensors, to obtain quantitative information on near UV aerosol absorption. Given the lack of observation-based data on ALH over such a long period, the proposed use of MERRA-2 generated data seems a reasonable approach.

The suggested approach implies the need to account for the multiple spectral dependencies of the UVAI on aerosol microphysical and optical properties, aerosol amount and its vertical distribution (ALH), as well as surface effects including land reflectance and ocean color effects.

The authors go over a series of mathematical definitions of ALH and settle on a particular method. In the next step, they carry out a radiative transfer-based analysis of UVAI sensitivity to their adopted ALH definition and use the results of such analysis to evaluate the suitability of existing ALH products.

**General Comments**

In their theoretical analysis, the authors underestimate the importance of many physical mechanisms that contribute to the observed spectral dependence captured by the measured UVAI. Their analysis disregards the well-known UVAI dependence on factors other than ALH such as the spectral dependence of the near UV imaginary component of aerosol particle refractive index, also expressed as Aerosol Absorption Exponent (AAE), and the spectral dependence of surface albedo which over arid and semi-arid regions of the world and over the oceans(pure water, chlorophyll and CDOM absorption) contribute significantly to the UVAI signal.

These effects are clearly non-negligible. For instance, at a particular viewing geometry, a 340-380 defined UVAI value of 2.0 for an aerosol layer of AOD (550 nm) 0.5, can be explained by multiple combination of ALH values between about 3 km and 6 km and AAE values between 1 and about 3. Thus, to properly derive ALH information, AAE must be accurately constrained. AAE vary significantly between aerosol types, and even for a given aerosol type the AAE varies regionally depending on soil composition for dust particles, and on fuel composition for carbonaceous particles.

The simplified radiative transfer calculations used in this analysis are based on unrealistic representation of the aerosol scattering phase function. The assumed Henvy-Grennstein (H-G) phase function used in this work does not allow the accurate modeling of the role particle size distribution, particle shape, and complex refractive index. These properties vary significantly for different aerosol types. Accurate radiative transfer calculations using Mie Theory for spherical particles and T-matrix and Geometric Optics combinations for non-spherical particles, must be used to reliably interpret the information content of satellite near UV observations. The aerosol model used in the analysis based on simplified assumptions on asymmetry factor and single scattering albedo is a very crude representation unsuitable for the analysis of satellite data.

For the above stated reasons, this manuscript is not publishable it its current form. A realistic representation of the relevant aerosol types as well as accurate radiative transfer calculations are required

to successfully extract the ALH information contained in the UVAI.

 Specific comments:

Line 32. The term 'small' is ambiguous. Refer to agree-upon size ranges in aerosol definition.

Line 45. Define columnar ALH.

Line 58. Since POLDER does not have any near UV channels, this is a rather strange statement.

Line 64. Clarify that EPIC does not involve spectrally resolved measurements.

Line 67. This is not a peer-reviewed work.

Line 71. The beginning of the paragraph refers to active measurements, but the following discussion abruptly changes to passive measurements. Thus, the whole paragraph looks incongruent.

Line 85. Altitude dependence is just one of the several dependencies of UVAI: spectral surface albedo and aerosol absorption exponent are very important. Discuss those effects and their importance in UVAI interpretation.

Line 114. The statement is not clear. Agreement of what with what.

Line 127. The validation shown in Appendix B uses CALIOP 532 nm extinction profiles. For smoke aerosols, 532 nm profiles are often truncated, and the derived aerosol height is overestimated [Torres et al., 2013] as a result of signal attenuation due to black carbon absorption (Kim et al., JGR, 118, 2013; Kacenelenbogen et al., JGR, 119, 2014; Liu et al., ACP, 15, 2015; Torres et al., AMT, 2013). Should use instead the 1064 nm channel.

Line 152. Aerosol effective geometric height makes more sense.

Line 255. The OMAERUV ALH climatology cannot be considered a single source ALH data base. This product is mainly based on a CALIOP climatology, but also includes model-based assumptions and spatial extrapolations developed to constrain the AOD/SSA retrieval. The authors should not use this data base. They should instead evaluate the CALIOP product on its own merits.

Line 283. Scattering effects of HG idealized aerosols differ significantly from real aerosols. HG aerosols is a 1950's tool when computing tools were inadequate and, no actual satellite observations were available.

Line 292. What is the point of using non-accessible data?

Line 296. Reference wavelength? How much aerosol is then rejected? At 550 nm only a small fraction of AOD is larger than 0.5. How much data is available after AOD lower than 0.5 is rejected?

Line 314. Same comment as above. HG aerosol type does not exist in the real world.

Line 321. This statement is not correct. CALIOP measures the actual aerosol vertical distribution.

Line 324. Need to explain better the diagnostic role of AOD.  AOD must be accurately known to retrieve realistic ALH. How accurate is the retrieved AOD, and how this accuracy translates in accuracy of retrieved ALH? It can be evaluated by comparison to ground-based observations or to MODIS-MiSR products.

Line 326-27. Not the same as O2-O2. It was stated above that AOD < 0.5 are discarded

Line 344. Please provide a peer-reviewed reference.

Line 349. Provide a validation reference like a comparison to CALIOP.

Line 351. This is another non-accessible data set.

Line 358. Retrieval capability over deserts is important. Please include a comparison to CALIOP over the Saharan Desert.

Line 380. Again HG. This sensitivity analysis must be carried out with actual radiative transfer calculations using Mie Theory for spherical particles and the adequate approach for non-spherical particles.

Line 383.How about in the UV? Spectrally invariant SSA in the UV s not realistic.

Line 389. The spectral dependence of surface albedo is important because it generates a 'spurious' aerosol signal. It may not be that important in single channel retrievals, but it is quite important for UVAI analyses. The high background UVAI (about 1.0) over arid regions is the result of surface albedo wavelength dependence in the near UV.

Line 393. The current UVAI definition in OMAERUV is not consistent with Equation 5. The new definition accounts for water cloud effects and surface spectral dependence. For historical reasons, the traditional definition (Equation 5) is also reported in the OMAERUV product under a different variable name. Authors should make sure they are using the correct parameter.

---

## Referee Comment (RC2) · Anonymous Referee #1 · 29 Apr 2020

The main focus of this paper is to assess if the aerosol layer height derived from the MERRA-2 aerosol vertical distributions is representative to the observed ALH that could provide long-term ALH retrospectively before the satellite retrieval of ALH is available. It also evaluates several ALH retrievals from OMI, TROPOMI, and GOME2. The evaluation matrix is the OMI UVAI that is a function of aerosol layer height and absorbing aerosol amount; if the ALH increases with the increase of UVAI and the magnitude of such increase is stronger with higher AOD, then the ALH product from either satellite or MERRA-2 is deemed robust or reliable. At the end, with five satellite products and four different definition of ALH calculated from the MERRA-2 vertical profiles, it concludes that products from TROPOMI O2-A band and GOME2 retrievals and the

MERRA-2 aerosol layer top height are the most suitable products for ALH.

General comments

I find this paper is interesting to show the differences between the ALH products and appreciate the thoroughness of the comparisons. I am also happy to see the MERRA-2 vertical profiles, which is largely simulated by a CTM, is robust enough to be representative of the ALH. On the other hand, I do have several concerns listed below that should be clarified/addressed before the paper is accepted for publication.

1. Different ALH products should be put into a context of the physical/optical meaning. As indicated in the paper, different products have different definitions. To help the readers grasp what they actually retrieve, it would be very helpful to have a reference dataset to indicate the vertical locations of these ALH. I think the CALIOP data can be used as such reference, i.e., the altitude of various ALH can be plotted on the CALIOP vertical curtain to show if the products represent the aerosol top, or the peak height, or optically weighted height, or something else.

2. It has been emphasized several times in the paper that the purpose is "to find an ALH data set for interpreting aerosol absorption from UVAI". I wonder how can that be achieved from a "robust" ALH? UVAI depends not only the ALH but also the amount of absorbing aerosol in the atmosphere. It will be helpful to add a few sentences how aerosol absorption (e.g., aerosol absorption optical depth) can be obtained from knowing the UVAI and ALH.

3. It is not clear to me if the feature of altitude-AOD dependence on UVAI or the actual altitude is more important for interpreting aerosol absorption from UVAI? For example, the study finds three products, TROPOMI O2-A, GOME2, and MERRA-2 Ht are the robust products judging from their features of altitude and AOD variation with UVAI. However, the actual altitudes are far apart among the three: taking from Figure 2 and 5 in the southern hemispheric biomass burning season, the altitudes are 2-2.5 km, >10 km, and ∼4 km for TROPOMI, GOME2 and MERRA-2 Ht over southern Africa (similar

discrepancy in South America as well). How will you use these remarkably different ALH in retrieving the aerosol absorption?

4. Using AERONET to "quality control" the ALH from OMAERUV over ocean: This is very unclear. The paper identified that OMAERUV ALH is extremely high (without approve) so additional quality control is necessary. However, 1) AERONET data is overwhelmingly over land and it has only a few sites on the islands, and 2) AERONET data does not have any ALH information to be used to "correct" OMAERUV ALH. In that regards, CALIOP would be more useful, but it is part of the climatology of ALH built in the OMAERUV already. Also, I found that dismissing the OMAERUV ALH data is unnecessary.

5. The slopes in some of the figures (e.g., Figure 7, 10) looks very strange – they don't go through the data points. Please confirm.

Specific comments:

Page 1, line 32: I would delete "occasionally". Tropospheric aerosols are being transported across the tropopause over the Asian summer monsoon region regularly every summer.

Page 2, line 51-52, lidar data: True, lidar data are limited in spatial or temporal coverages, but for this work the statistics is the most important, not event-by-event. I strongly encourage to use the lidar data such as CALIOP, since you already used them for MERRA-2 evaluation.

Page 2, line 65-66 to Page 3, line 67: The sentence is recursive: AAH has become an official product of the GOME-2…that retrieves the AAH. Revise.

Page 3, line 82: How good is "good"? e.g., within x meters? within y%? To what extend the error is acceptable?

Page 4, MERRA-2: Just keep that in mind that MERRA-2 aerosol vertical profile is NOT a reanalysis product. Only column AOD is.

Page 4, line 128: Again, please avoid the subjective adjectives such as "good". Be more quantitative.

Page 7, line 181-182: 366 days in 2016 x 100 profiles/day = 36600 profiles. Where is 401800 coming from?

Page 7, line 183: How did you change the vertical grid size in MERRA-2?

Page 7, line 187: I think the vertical grid is also denser near the tropopause in MERRA-2.

Page 8, line 214-215, "…Ht is better associated with AOD", and Figure 2: I wonder why anyone should expect ALH to be positively associated with AOD. The common knowledge is that the transported plumes are lifted higher (i.e., higher Ht) but AOD is lower than those near the source region. In that regards, Ht makes less sense. It would be informative if you could plot the profiles over the source regions as well as over the transported regions (e.g., North Pacific, North Atlantic) for comparison. Figure 4 only shows the vertical profiles over the source regions. Profiles over open ocean will be helpful.

Page 9, line 271: "extreme" high ALH in Figure 5a over oceans: it should be "extremely". How high is extremely high? Over ocean the ALH from OMAERUV is about 4 km. How does it compare with independent data?

Page 9, line 273: How is the over-ocean ALH determined in OMAERUV? Is most of them from CALIOP or CTM, or assumed according to aerosol type (thus may have high bias if UVAI is high-biased)? Besides, Fig 5b shows much lower UVAI over ocean than over land; how does the lower UVAI produce higher ALH over ocean?

Page 10, line 275-276, using AERONET data: I already stated that in my general comments.

Page 10, line 300+, and Figure 5: Fig 5d and 5g shows "extremely" high values of ALH with a good fraction of points above 10 km! Also, it seems that OMI O2O2 (5d-f) and

TROPOMI O2A (5g-i) are almost identical? This is a bit puzzling given the differences in retrieving methods, pixel resolution, and spatial coverage (e.g., row anomaly of OMI).

General comment for Section 3: It would be helpful to indicate the nature of these different ALH products from satellite retrievals, for example, it is more representative of top of the aerosol layer, or median, or optical centroid, or effective (give definition), etc.

Page 12, line 366: What is CMT – typo?

Page 14, line 418: "...are less variable" – I would say "...have little variation".

Page 15, line 428-429: Why are the AOD thresholds different for different ALH products?

Page 15, paragraph starting at line 30: The AERONET quality screen is only applied to the OMAERUV data over the ocean, right?

Page 15, line 434-435: This needs to be quantified - if you use the original OMAERUV data without AERONET screening, how much difference does it make?

Page 15, line 449 and 457: the products and websites are not consistent, e.g., product is MYD08_D3 but the website is MOD08_M3 in line 449.

Page 16, line 479: I don't understand the argument that R > 0.6 for OMAERUV is unrealistic. The ALH in OMAERUV is mainly from CALIOP climatology, or a CTM; the UVAI is only used for selecting aerosol type. And it is not clear how much help the screening with AERONET data helps for a better correlation; if this is the reason to not exclude the OMAERUV data, then just don't apply the screening.

---

## Author Comment (AC1) · 27 May 2020

**#Response to the RC1**

**General Comments**

In their theoretical analysis, the authors underestimate the importance of many physical mechanisms that contribute to the observed spectral dependence captured by the measured UVAI. Their analysis disregards the well-known UVAI dependence on factors other than ALH such as the spectral dependence of the near UV imaginary component of aerosol particle refractive index, also expressed as Aerosol Absorption Exponent (AAE), and the spectral dependence of surface albedo which over arid and semi-arid regions of the world and over the oceans(pure water, chlorophyll and CDOM absorption) contribute significantly to the UVAI signal.

These effects are clearly non-negligible. For instance, at a particular viewing geometry, a 340-380 defined UVAI value of 2.0 for an aerosol layer of AOD (550 nm) 0.5, can be explained by multiple combination of ALH values between about 3 km and 6 km and AAE values between 1 and about 3. Thus, to properly derive ALH information, AAE must be accurately constrained. AAE vary significantly between aerosol types, and even for a given aerosol type the AAE varies regionally depending on soil composition for dust particles, and on fuel composition for carbonaceous particles.

The simplified radiative transfer calculations used in this analysis are based on unrealistic representation of the aerosol scattering phase function. The assumed Henvy-Grennstein (H-G) phase function used in this work does not allow the accurate modeling of the role particle size distribution, particle shape, and complex refractive index. These properties vary significantly for different aerosol types. Accurate radiative transfer calculations using Mie Theory for spherical particles and T-matrix and Geometric Optics combinations for non-spherical particles, must be used to reliably interpret the information content of satellite near UV observations. The aerosol model used in the analysis based on simplified assumptions on asymmetry factor and single scattering albedo is a very crude representation unsuitable for the analysis of satellite data.

For the above stated reasons, this manuscript is not publishable it its current form. A realistic representation of the relevant aerosol types as well as accurate radiative transfer calculations are required to successfully extract the ALH information contained in the UVAI.

**Author's response:**

Thank you for your suggestion! UVAI is a qualitatively derived from level 1 radiance measurements. It depends on many aspects, among which the most important are the aerosol loading, aerosol vertical distribution and aerosol absorption. In this paper, we aim to find / build up an ALH data set that we focus on the UVAI's dependence on aerosol vertical distribution.

But there are two misunderstandings: (1) we are **not** using radiative transfer-based analysis of UVAI sensitivity to ALH. All the ALH-UVAI relationship analysis in this paper is only statistically based rather than physically based. In other words, we directly use the existing satellite UVAI product and co-located to the corresponding ALH data. For the MERRA-2 derived ALH variables (defined by us), the OMAERUV UVAI is co-located to the MERRA-2 and then make the analysis. For other satellite data sets (OMI, TROPOMI and GOME-2), the UVAI is combined with their corresponding ALH. (2) the radiative transfer simulations in this paper is **only used for sensitivity study** to present to provide some intuitions for readers who are not familiar with UVAI that how the UVAI is response to the aerosol vertical distribution. We have to admit that the setup of the sensitivity study is simplified. But as we will show, even using the Mie scattering scheme, wavelength dependent surface albedo and aerosol absorption, the conclusion will not change significantly. The HG aerosol models mentioned in other cases, are used in the (either official or not) ALH retrieval algorithms. Please see responses to specific comments for more details.

**Specific comments: Line 32. The term 'small' is ambiguous. Refer to agree-upon size ranges in aerosol definition. **Author's response:** Corresponding change has been done. **Change of the manuscript:** Atmospheric aerosols are liquid or solid particles with typical particle size ranging from $10^{-4}$ to $10 \ \mu m$ that originate from natural or anthropogenic sources.**

Line 45. Define columnar ALH. Author's response: Corresponding change has been done.

**Change of the manuscript:**

ALH is a compact representation of aerosol profiles with a single value.

**Line 58. Since POLDER does not have any near UV channels, this is a rather strange statement.**

Author's response:

Corresponding change has been done. **Change of the manuscript:**

Change of the manuscript:

The POLarization and Directionality of the Earth's Reflectance (POLDER) onboard PARASOL utilizes the distinct polarization difference between air molecules and aerosol particles (Dubovik et al., 2011).

**Line 64. Clarify that EPIC does not involve spectrally resolved measurements.**

Author's response:

EPIC does observe in several spectral bands, including in the O2 A and B bands.

Line 67. This is not a peer-reviewed work.

Author's response:

The ATBD is reviewed and contains useful information to readers. It is an ATBD of GOME-2 absorption aerosol layer height product that will be released soon.

**Line 71. The beginning of the paragraph refers to active measurements, but the following discussion abruptly changes to passive measurements. Thus, the whole paragraph looks incongruent. **Author's response:**

It is a matter of layout issue. They just look like a single paragraph but actually is not.

The paragraph about active measurements starts from "Aerosol vertical distributions are either described by..." to "...missing data in the measured profiles". The paragraph about passive measurements starts with "ALH is usually retrieved from..." to the end.

**Line 85. Altitude dependence is just one of the several dependencies of UVAI: spectral surface albedo and aerosol absorption exponent are very important. Discuss those effects and their importance in UVAI interpretation. Author's response:**

As declared at the beginning of this document, we are aware of that the UVAI definitely only depends on ALH, but also other parameters as you mentioned. The UVAI's introduction and its dependence are detailed described later in section 4.1.

Line 114. The statement is not clear. Agreement of what with what.

Author's response: Corresponding change has been done. Change of the manuscript: Improved agreement between MERRA-2 and observations is also found for aerosol optical properties and aerosol vertical distributions (Buchard et al., 2017)

Line 127. The validation shown in Appendix B uses CALIOP 532 nm extinction profiles. For smoke aerosols, 532 nm profiles are often truncated, and the derived aerosol height is overestimated [Torres et al., 2013] as a result of signal attenuation due to black carbon absorption (Kim et al., JGR, 118, 2013; Kacenelenbogen et al., JGR, 119, 2014; Liu et al., ACP, 15, 2015; Torres et al., AMT, 2013). Should use instead the 1064 nm channel. **Author's response:**

It is true that lidar backscattering coefficient attenuation at 532 nm is stronger than 1064 nm. But we insist using profiles at 532 nm for the following concerns: (1) we are comparing CALIOP and MERRA-2, while the latter only report aerosol optical properties at 550 nm. Lidar aerosol properties is closer to compare; (2) the loss of measurement sensitivity near surface may affect the ALH determination, but here we are comparing the profiles rather than ALH. Besides, the common extinction weighed ALH suffers the problem of the lower sensitivity near surface, but for aerosol layer top height, its influence is small. Finding a representative ALH is one of the aims of this paper; (3) CALIOP level 3 monthly data does not provide profiles at 1064 nm. We use level 3 data as the time period is 2006-onwards in this study.

We will add this as a notice in the manuscript. Change of the manuscript:

As the climatology reports aerosol optical properties at 532 nm only, one should keep in mind that extinction coefficient at visible has lower measurement sensitivity in the lower part of the atmosphere due to strong attenuation by smoke (Torres et al., 2013, Kim et al., 2015).

Line 152. Aerosol effective geometric height makes more sense. **Author's response:** Corresponding change has been done. **Change of the manuscript:** 2.2.3 Aerosol effective geometric height

Line 255. The OMAERUV ALH climatology cannot be considered a single source ALH data base. This product is mainly based on a CALIOP climatology, but also includes model-based assumptions and spatial extrapolations developed to constrain the AOD/SSA retrieval. The authors should not use this data base. They should instead evaluate the CALIOP product on its own merits.

**Author's response:**

We have noticed that the OMAERUV ALH is not an operational product, as we discussed in section 3.1. The reason we still include it in this paper is because that the OMAERUV ALH is designed to retrieve accurate aerosol properties of absorbing aerosols in the UV channel (Torres et al., 2013), and it has a long-term global record since 2006. Furthermore, we want to use the relationship between UVAI and ALH that is found in the sensitivity study (section 4.1) to analytically examine whether the OMAERUV ALH and its corresponding UVAI has this relationship (and also for other satellite observations). If the UVAI-ALH relationship is found, then no matter the ALH is from observation retrievals or model simulations, we will consider it is worth analyzing UVAI.

The result is that the OMAERUV UVAI is highly dependent to OMAERUV ALH, even though we know this relationship is too 'optimistic', because the ALH is assigned according to aerosol types, the latter is (partly) determined by UVAI (section 4.4).

The CALIOP data, however, facing the problem we discussed in section 1 Introduction. Given a profile, which definition should be used to derive an ALH? On the other side, the CALIOP is subject to the signal attenuation due to presence of heavy clouds or aerosols, leading to missing data in profiles (incomplete profiles), which adds a lot of uncertainties on the retrieved ALH. Also, the coverage of CALIOP data is much less than that of OMI.

Thus, instead of using CALIOP, we use the AERONET to assure the quality of OMAERUV ALH by co-locating the two data sets and applying quality constraints on AOD and SSA retrievals as described in section 4.2.

**Line 283. Scattering effects of HG idealized aerosols differ significantly from real aerosols. HG aerosols is a 1950's tool when computing tools were inadequate and, no actual satellite observations were available. Author's response:**

It is the aerosol models defined by Chimot et al. (2017) to retrieve ALH over O2-O2 band. Although the aerosol properties are simplified, but the HG is able to reproduce the Mie scattering function reasonably well for most aerosol types (Chimot et al., 2017). The HG assumptions is also applied in Oxygen-A band ALH retrieval (Sander et al., 2015) and aerosol corrections in AMF calculation when retrieving trace gases (Spada et al., 2006; Wagner et al., 2007; Castellanos et al., 2015).

The reason that HG is often used in the O2-A band retrieval is still computational efficiency. Even if the single HG phase function is replaced by a single Mie scattering phase function, this will increase computational effort significantly. Note that for example the TROPOMI ALH radiative transfer is based on a dataset that takes several months to produce on a computer cluster.

**Line 292. What is the point of using non-accessible data?**

**Author's response:**

Because the OMI O2-O2 ALH is not an official level 2 product, although Chimot et al. are making efforts on that. If interested in the data, one can contact with Chimot directly. We have mentioned this in section Data availability.

**Line 296. Reference wavelength? How much aerosol is then rejected? At 550 nm only a small fraction of AOD is larger than 0.5. How much data is available after AOD lower than 0.5 is rejected? **Author's response:**

The 0.5 threshold applied to AOD is at 550 nm. About 2% of data is available after AOD lower than 0,5 is rejected. It is a quality control suggested by Julien Chimot, who is mainly in charge of the OMI O2-O2 ALH data set. As we mentioned in the manuscript, the aerosol shielding effect on O2-O2 absorption is too low for low AOD cases. Nevertheless, even though only 2% data left, there are still around 50.000 samples left to analysis, which is statistically sufficient.

**Line 314. Same comment as above. HG aerosol type does not exist in the real world.**

**Author's response:**

It is the aerosol models defined in Sander et al. (2015) and Sander and de Haan (2016) for the official TROPOMI Oxygen A-band ALH product. Same explanation as previous.

**Line 321. This statement is not correct. CALIOP measures the actual aerosol vertical distribution. Author's response:**

CALIOP measures the actual aerosol vertical distribution. But due to the presence of heavy clouds and aerosols, the lidar signal tends to attenuate, which may lead to missing data in the measured profiles. Considering that CALIOP lidar signal passes from the top of the atmosphere to the surface, the sensitivity should be higher at upper clean atmosphere than that passes through clouds or aerosol layers.

**Line 324. Need to explain better the diagnostic role of AOD. AOD must be accurately known to retrieve realistic ALH. How accurate is the retrieved AOD, and how this accuracy translates in accuracy of retrieved ALH? It can be evaluated by comparison to ground-based observations or to MODIS-MiSR products.**

**Author's response: s**

The TROPOMI ALH algorithm is an iterative optimal estimation scheme that fits the ALH and an effective AOD. The AOD is an effective quantity, that is based on single phase function assumption. As such, it is treated as a byproduct of the retrieval, which because of the use of a single-phase function is not optimized to be close to the truth. A typical example of the effective AOD is given in the figure below, comparing the ALH AOD at 758 nm with the NPP-VIIRS AOD at 550 nm for a Saharan dust plume (note that the color scales are optimized to show the spatial similarity over the ocean). This figure shows that the spatial correlation of the two products is very high. Over land the ALH AOD shows overestimates, which is due to the high surface albedo in this region and limited accuracy of the used albedo climatology, which makes the ALH retrieval over such high surface albedo regions very challenging.

Left Panel: Effective AOD from the TROPOMI ALH retrieval at 758 nm. Right panel AOD from NPP-VIIRS dark target retrieval at 550 nm. Data are off the coast of Africa for 2020-05-02.

In this manuscript, we use the ALH data that are quality filtered according to the recommendations. It is therefore beyond the scope of this manuscript to perform a full validation of the effective AOD in the ALH product. A validation of the ALH product is described in Nanda et al. (2019, AMTD). The TROPOMI ALH generally can

captures the aerosol layers presented in CALIOP extinction coefficient profiles. CALIOP ALH is generally higher than TROPOMI by 1-2.4 km (but it also depends on how you calculate the ALH, in their work, they determined the CALIOP ALH as extinction coefficient weighed height).

**Line 326-27. Not the same as O2-O2. It was stated above that AOD**

---

## Author Comment (AC2) · 27 May 2020

**Response to the RC2**

General comments
I find this paper is interesting to show the differences between the ALH products and
appreciate the thoroughness of the comparisons. I am also happy to see the MERRA-2
vertical profiles, which is largely simulated by a CTM, is robust enough to be representative
of the ALH. On the other hand, I do have several concerns listed below that
should be clarified/addressed before the paper is accepted for publication.

1. Different ALH products should be put into a context of the physical/optical meaning.
As indicated in the paper, different products have different definitions. To help the
readers grasp what they actually retrieve, it would be very helpful to have a reference
dataset to indicate the vertical locations of these ALH. I think the CALIOP data can be
used as such reference, i.e., the altitude of various ALH can be plotted on the CALIOP
vertical curtain to show if the products represent the aerosol top, or the peak height, or
optically weighted height, or something else.
**Author's response:**
It is a good advice. But we have to stress that exploring what is the physical/optical meaning of each ALH products
is not the main purpose of this work (we are aiming to find an ALH that follows our understanding of UVAI altitude
dependence). Instead, we will add the comparison between various ALH product with CALIOP climatology in
Appendix D.
**Change of the manuscript:**
Appendix D: validating satellite ALH with CALIOP measurements
We validate satellite ALH products used in this paper with CALIOP measurements. Fig. D1 presents the zonal
average of each ALH product (blue and green lines) against corresponding CALIOP climatology (contour). For
better understanding the physical mean of each ALH products, we also calculate the ALH from CALIOP extinction
profiles using ALH definitions introduced in Section 2.
The ALH reported in OMAERUV (first column) generally presents lower values except for that at high latitudes and
the mid latitude in the Southern Hemisphere. This is consistent with Fig .5a, where OMAERUV ALH over remote
ocean is significantly higher than source regions. The potential reasons, as we explained in Section 3.1, are aerosol
type misclassifications, the unrealistic ALH assumptions when no climatology data available, and high sensitivity to
outliers due to less observations, etc. Between 10-30°N where the dominant aerosols come from the world dust belt
according to Fig.5c, the OMAERUV ALH matches well with the 'effective' heights ($H_{aer}^{\beta}$, $H_{aer}^{\tau}$ and $H_{aer}^{63}$) derived
from CALIOP climatology, which may own to the proper ALH determination for dust aerosols (Torres et al., 2013).
However, the ALH is almost close to the surface at biomass burning regions in tropics (0-30°S). The reason could
be that the ALH for smoke aerosols is determined to be less than 3 km if no climatology data is available (Torres et
al., 2013).
The comparison between OMI O2-O2 ALH and CALIOP (second column) only covers 3 seasons as CALIOP data
is only available from June 2006. The ALH retrieved by SSA = 0.90 (blue line) is slightly lower than that retrieved
by SSA = 0.95 (green line), which agrees with conclusions in Chimot et al. (2017). The OMI O2-O2 ALHs are close
to the top boundary of the aerosol layer ($H_{aer}^{t}$) within 0-30°S. Combined with data distribution provided in Fig.5d-f,
the ALH well presents the smoke plumes over tropics and subtropics. It also captures the dust outbreaks during JJA.
During DJF and SON, the ALHs are retrieved mainly for urban aerosol in East Asia, where aerosol extinction is
strongest near surface. The magnitude of OMI O2-O2 ALH is between the effective height and the top boundary
height. However, in high latitude in the Southern Hemisphere, the ALH is remarkedly high. The reason is that the
high sensitivity to the retrieval outliers due to the lack of retrievals (Fig.5d, 5g).
The last two column shows the comparison between TROPOMI ALH and GOME-2 AAH with CALIOP profiles.
Although the TROPOMI ALH and GOME-2 AAH are collected during the similar period, the latter has larger
variations due to the wider spatial coverage over both continents and oceans (Fig.5l, 5o). Similar to the OMAERUV
ALH, the magnitude of both TROPOMI ALH and GOME-2 AAH match well with the 'effective' heights ($H_{aer}^{\beta}$,
$H_{aer}^{\tau}$ and $H_{aer}^{63}$) derived from the CALIOP profiles at latitudes between 10 and 30°N. At these locations, both ALH
products are mainly retrieved for dust outflows over ocean. Disagreements between these two satellite ALH
products and CLAIOP profiles appear at mid to high latitudes regions. At these regions, the satellite ALHs are only
available for individual cases (with UVAI larger than certain values) that are not visible in CALIOP profiles due to

averaging effects. For example, the high values of TROPOMI ALH present the smoke plumes due to fire events at the South America in DJF, Australia in DJF and MAM and South Africa in SON, and the high values of GOME-2 AAH present the smoke plumes due to fire seasons in the North America from MAM to SON and Asia in MAM and JJA. The Australia fire presented in TROPOMI ALH but invisible in GOME-2 AAH is also due to the averaging effect, since the latter has wider coverage over ocean (AAH over remote ocean is generally lower compared with smoke plumes).

In summary, the OMAERUV ALH, TROPOMI ALH and GOME-2 AAH indicate the 'effective' height (the altitude where aerosol extinction is strongest) for dust dominant aerosols layers. Large discrepancy between these satellite ALHs and CALIOP profiles are found at mid and high latitudes. For OMAERUV ALH, the disagreement mainly due to the unreasonable ALH determination, while for the other two products, the mismatches also result from the difference in spatial coverages due to the retrieval algorithm limitation and averaging effects of CALIOP profiles. On the other hand, the OMI O2-O2 ALH varies along with CALIOP profiles and captures the aerosol layer top boundary height for smoke aerosols, although it is sensitive to outliers in high latitude regions.

[Figure]

**Figure. D1 The zonal average and the standard deviation of the satellite ALHs (blue lines) against the zonal average of the CALIOP extinction coefficients (contours) during the corresponding periods. Rows represent different seasons and columns represent different ALH products. For OMI O2-O2 ALH (second column), the ALHs retrieved SSA = 0.90 and 0.95 are indicated by blue and green line, respectively. The ALHs calculated for CALIOP profiles using different ALH definitions introduced in Section 2 are also indicated by different markers.**

2. It has been emphasized several times in the paper that the purpose is "to find an ALH data set for interpreting aerosol absorption from UVAI". I wonder how can that be achieved from a "robust" ALH? UVAI depends not only the ALH but also the amount of absorbing aerosol in the atmosphere. It will be helpful to add a few sentences how aerosol absorption (e.g., aerosol absorption optical depth) can be obtained from knowing the UVAI and ALH.

**Author's response:**

The motivation of this paper is that, UVAI is a qualitative measure of aerosol absorption. It is a global data sets with long-term record since 1978, which is of potential to derive quantitative aerosol absorption, such as SSA or AAOD. In our previous studies (Sun et al., 2018, 2019), we have proved this possibility for individual cases. But to have a global aerosol absorption database, one of the major challenging is the lack of aerosol vertical distribution data. The aim of this paper that finds such an ALH data set for interpreting aerosol absorption from UVAI. This we used to put in the introduction part. Here we have made some changes to make it clearer.

**Change of the manuscript:**

From our perspective, quantitatively interpretation of aerosol absorption from the ultra-violet aerosol index (UVAI) satellite records requires information on the aerosol vertical distributions. No matter deriving SSA by forward radiative simulations of UVAI (Sun et al., 2018) or by statistically based machine learning methods(Sun et al., 2019), the aerosol vertical distribution is always one of the important factors to UVAI. So far, UVAI has a fourdecade global record, while a corresponding long-term global daily aerosol vertical distribution database is not yet available. (line 46-51)

3. It is not clear to me if the feature of altitude-AOD dependence on UVAI or the actual altitude is more important for interpreting aerosol absorption from UVAI? For example, the study finds three products, TROPOMI O2-A, GOME2, and MERRA-2 Ht are the robust products judging from their features of altitude and AOD variation with UVAI. However, the actual altitudes are far apart among the three: taking from Figure 2 and 5 in the southern hemispheric biomass burning season, the altitudes are 2-2.5 km, >10 km, and _4 km for TROPOMI, GOME2 and MERRA-2 Ht over southern Africa (similar discrepancy in South America as well). How will you use these remarkably different ALH in retrieving the aerosol absorption?

**Author's response:**
As we discussed in section 1 Introduction, different measuring techniques, different retrieval algorithms, and different assumptions made in retrieval algorithms, different definitions of ALH, etc. make it difficult to directly compare different ALHs. Also as described in Torres et al. (2013), a direct comparison between above ALH data sets is not very meaningful as these definitions represent different aspects of the aerosol vertical distribution (section 4).
Alternatively, we focus on the UVAI dependence on ALH for two reason: (1) this relationship is comparable among different data sets; (2) we are interested in the relationship rather than the absolute ALH values, as ALH itself is just a representation of aerosol profiles, rather than the actual location of aerosol layers. When we are using machine learning methods to derive aerosol absorption from UVAI, the UVAI-ALH relationship is more important than their absolute values, as input features are usually normalized (so the absolute values do not matter).

4. Using AERONET to "quality control" the ALH from OMAERUV over ocean: This is very unclear. The paper identified that OMAERUV ALH is extremely high (without approve) so additional quality control is necessary. However, 1) AERONET data is overwhelmingly over land and it has only a few sites on the islands, and 2) AERONET data does not have any ALH information to be used to "correct" OMAERUV ALH. In that regards, CALIOP would be more useful, but it is part of the climatology of ALH built in the OMAERUV already. Also, I found that dismissing the OMAERUV ALH data is unnecessary.

**Author's response:**
Note that the AERONET data screening is applied to both AOD and SSA of OMAERUV and MERRA-2, i.e. the AOD and SSA of OMAERUV and AOD and SSA of MERRA-2 should pass the criteria at the same time, then the pixel is accepted, otherwise, rejected. We apply this screening for two reasons:
(1) to improve the data quality. The OMAERUV ALH (SSA) is not a real retrieved parameter but a 'by-product' during the AOD retrieval. The AERONET does not containing information on aerosol vertical distribution, but by constraining the OMAERUV AOD as well as SSA quality using AERONET measurements, we expect a better quality of ALH, although it may not be always true. As shown in the following figures, the data after AERONET screening is indeed better than the raw data at least in terms of UVAI-ALH relationship. For the same reason, the AERONET quality screening is also applied to the MERRA-2. Although MERRA-2 AOD is partly assimilated from AERONET, the SSA is not;
(2) to directly compare the UVAI-ALH relationship of OMAERUV and MERRA-2. Using AERONET screening ensures that OMAERUV and MERRA-2 are reference to the same AERONET samples, thus they can compare to each other under the same condition. This method is not applicable to other data sets (OMI O2-O2, TROPOMI and GOME2) as the data availability is only near 1 year in this study, the number of collocated data (OMAERUV-OMIO2O2/TROPOMI/GOME-2-AERONET) is not statistically representative.

We admit that the AERONET screened data suffers from the even coverage over land and oceans, but the AERONET is still the first choice for validating satellite retrieved aerosol properties. The correction of OMAERUV ALH using near-real time CALIOP is actually not necessary, as we want to evaluate the UVAI-ALH relationship of OMAERUV.

[Figure]

5. The slopes in some of the figures (e.g., Figure 7, 10) looks very strange – they don't go through the data points. Please confirm.

**Author's response:**
The slopes are correct, they seem not go through the data with higher UVAI, because the data distribution is skewed to lower UVAI. In other words, there are many samples with smaller UVAI at the bottom of the plot which bias the slope to smaller values. It would be clearer on a density plot, but to show the information on AOD, we still use the scattering plot.

Specific comments:
Page 1, line 32: I would delete "occasionally". Tropospheric aerosols are being transported across the tropopause over the Asian summer monsoon region regularly every summer.

**Author's response:**
Corresponding change has been done.

**Change of the manuscript:**
Although initially emitted in the lower part of the atmosphere, aerosol particles can be transported across the tropopause and stay in stratosphere for several months (Islam et al., 2017).

Page 2, line 51-52, lidar data: True, lidar data are limited in spatial or temporal coverages, but for this work the statistics is the most important, not event-by-event. I strongly encourage to use the lidar data such as CALIOP, since you already used them for MERRA-2 evaluation.

**Author's response:**
Lidar data like CALIOP is more suitable for case evaluations, as long as there exist co-located samples. The MERRA-2 aerosol vertical distribution validation in Appendix B is based on monthly climatology.

As you suggested, we have added a comparison of various ALH data sets using CALIOP profiles as the reference in Appendix D, see previous changes.

Page 2, line 65-66 to Page 3, line 67: The sentence is recursive: AAH has become an official product of the GOME-2: : :that retrieves the AAH. Revise.

**Author's response:**
Corresponding change has been done.
**Change of the manuscript:**
Recently, the absorbing aerosol layer height (AAH) has been developed to be an official product of the GOME-2 instrument (Tilstra et al., 2019).

Page 3, line 82: How good is "good"? e.g., within x meters? within y%? To what extend the error is acceptable?
**Author's response:**
The validation is not based on ALH but extinction profiles. According to Buchard et al. (2017), they validate the seasonal MERRA-2 aerosol profiles with CALIOP for regions of interests. Compared with M2REPLAY (control run without AOD assimilation), the MERRA-2 profiles show similar structure as CALIOP, with maximum attenuated backscatter at about the same height. MERRA-2 tends to underestimate sea salt backscatter attenuation over ocean. This is mentioned in the corresponding MERRA-2 section (section 2.1).

Page 4, MERRA-2: Just keep that in mind that MERRA-2 aerosol vertical profile is NOT a reanalysis product. Only column AOD is.
**Author's response:**
Thank you for reminding, we have noticed that.

Page 4, line 128: Again, please avoid the subjective adjectives such as "good". Be more quantitative.
**Author's response:**
Same as previous response, the comparison is based on seasonal profiles rather than ALH.

Page 7, line 181-182: 366 days in 2016 x 100 profiles/day = 36600 profiles. Where is 401800 coming from?
**Author's response:**
We use MERRA-2 data from **2006** to **2016**, in total 11 years. 11 * 365 * 100 ~ 400000 profiles.

Page 7, line 183: How did you change the vertical grid size in MERRA-2?
**Author's response:**
We change the grid size by resampling the given profile in different resolution. We add more description on this point.
**Change of the manuscript:**
For each profile, we modify the profile by changing its vertical grid size (i.e. resample in different vertical resolution)…

Page 7, line 187: I think the vertical grid is also denser near the tropopause in MERRA-2.
**Author's response:**
The denser grid at the lower part of the atmosphere is relative to the upper part of the atmosphere. The vertical resolution is decreasing along the increasing altitude.

Page 8, line 214-215, ": : :Ht is better associated with AOD", and Figure 2: I wonder why anyone should expect ALH to be positively associated with AOD. The common knowledge is that the transported plumes are lifted higher (i.e., higher Ht) but AOD is lower than those near the source region. In that regards, Ht makes less sense. It would be informative if you could plot the profiles over the source regions as well as over the transported regions (e.g., North Pacific, North Atlantic) for comparison. Figure 4 only shows the vertical profiles over the source regions. Profiles over open ocean will be helpful.
**Author's response:**

Agree, not expectation that ALH is positivity associated with AOD. It is just that the Ht has this property because the Ht capture a certain threshold of extinction coefficient, where extinction above this threshold makes little contribution to the total AOD (the threshold varies with different profiles, i.e. inconstant). Corresponding change has been done.

But the lifted plume is not necessarily to have lower AOD than that aerosol layers near ground.

The Antarctica in Fig.4 is actually a non-source region (clean region). Fig.3 zonal average plot also contains profiles of source and non-source regions.

**Change of the manuscript:**

By contrast, the spatial-temporal variation of $H_{aer}^t$ is associated with AOD. One can easily recognize the seasonal aerosol sources from the spatial and temporal variation of $H_{aer}^t$, e.g. the biomass burning regions in the central Africa during winter, the Sahara dust and its outflows over the Northern Atlantic during summer, etc. It is because that the $H_{aer}^t$ captures the height above which the extinction coefficients make little contribution to the total AOD.

Page 9, line 271: "extreme" high ALH in Figure 5a over oceans: it should be "extremely".
How high is extremely high? Over ocean the ALH from OMAERUV is about
4 km. How does it compare with independent data?

**Author's response:**

The extremely high is respect to OMAERUV ALH at source regions. The OMAERUV ALH over (remote) ocean is around 4 km, while that over source regions is only around 2-3 km, which is not consistent with our knowledge. The direct comparison with independent data is not very meaningful as the ALH is differently defined. However, compared with GOME-2 AAH that ALH over open oceans is generally lower than that over source regions (OMI O2-O2 ALH and TROPOMI O2-A band ALH do not have many measurements over open ocean), the OMAERUV ALH over ocean seems extremely high.

We have made this statement clearer.

**Change of the manuscript:**

The extremely high ALH over open oceans relative to that over source regions (Fig.5a) may be caused by many factors.

Page 9, line 273: How is the over-ocean ALH determined in OMAERUV? Is most of them from CALIOP or CTM, or assumed according to aerosol type (thus may have high bias if UVAI is high-biased)? Besides, Fig 5b shows much lower UVAI over ocean than over land; how does the lower UVAI produce higher ALH over ocean?

**Author's response:**

The OMAEUV ALH determination starts with aerosol type classification, no matter there are CALIOP or CTM climatology available. Thus, the error in UVAI may lead to errors in ALH. But please be noted that it is not only the UVAI, but also the CO index (COI, calculated from ARIS) and the scene type is also used to determine the aerosol types.

**Change of the manuscript:**

The extremely high ALH over open oceans relative to that over source regions (Fig.5a) may be caused by many factors. For instance, the row anomalies and sun-glint pixels that cannot be detect by the quality flag may lead to errors in UVAI, and also errors in other support information for aerosol classification (carbon monoxide index and scene types) which further affects ALH determinations; the unrealistic a-priori assumptions of ALH when no climatological entry exists; or the high sensitivity to outliers as the number of observations is small over remote oceans, etc.

Page 10, line 275-276, using AERONET data: I already stated that in my general comments.

**Author's response:**

Please see the previous comment. The AERONET is still necessary to ensure the quality of OMAERUV ALH.

Page 10, line 300+, and Figure 5: Fig 5d and 5g shows "extremely" high values of ALH with a good fraction of points above 10 km! Also, it seems that OMI O2O2 (5d-f) and TROPOMI O2A (5g-i) are almost identical? This is a bit puzzling given the differences in retrieving methods, pixel resolution, and spatial coverage (e.g., row anomaly of OMI).

General comment for Section 3: It would be helpful to indicate the nature of these different ALH products from satellite retrievals, for example, it is more representative of top of the aerosol layer, or median, or optical centroid, or effective (give definition), etc.

**Author's response:**
The extremely high values in Fig.5d and g maybe due to the algorithmic errors.
The Fig.5d-i are OMI O2-O2 ALH (row 2-3). Fig.5d-f is OMI O2-O2 ALH retrieval by assuming SSA is 0.9 and Fig.5g-I is OMI O2-O2 ALH retrieval by assuming SSA is 0.95 in the NN-algorithm. Thus, they are almost identical. Please refer to section 3.2 for more details.
The comparison between each ALH products and CALIOP profiles is provided in Appendix D, see previous response.
**Change of the manuscript:**

Page 12, line 366: What is CMT – typo?
**Author's response:**
It is a typo. Corresponding change has been done.
**Change of the manuscript:**
…as CTMs do not have corresponding UVAI fields…

Page 14, line 418: ": : :are less variable" – I would say ": : :have little variation".
**Author's response:**
Corresponding change has been done.
**Change of the manuscript:**
Although the UVAI varies from 0 to 5, the effective heights have little variation in the highest AOD regime.

Page 15, line 428-429: Why are the AOD thresholds different for different ALH products?
**Author's response:**
Because different ALH data is retrieved / analyzed by different AOD data. OMAERUV ALH uses AOD at 500 nm from the same product. OMI O2-O2 ALH is retrieved based on MODIS data. TROPOMI ALH and GOME-2 AAH product does not have corresponding AOD retrieved, thus we use the co-located MODIS AOD to analyze.

Page 15, paragraph starting at line 30: The AERONET quality screen is only applied to the OMAERUV data over the ocean, right?
**Author's response:**
I think you are mentioning continent?
The comment on the AERONET data screening is mentioned in the response to the general comment.

Page 15, line 434-435: This needs to be quantified - if you use the original OMAERUV data without AERONET screening, how much difference does it make?
**Author's response:**
If no AERONET screening, the UVAI dependence on ALH becomes weaker. The AERONET screening has two roles, as explained in the beginning of this document.

Page 15, line 449 and 457: the products and websites are not consistent, e.g., product is MYD08_D3 but the website is MOD08_M3 in line 449.
**Author's response:**
Thank you for the correction. Corresponding change has been done.
**Change of the manuscript:**
The data access doi has changed into: http://dx.doi.org/10.5067/MODIS/MYD08_D3.006 and http://dx.doi.org/10.5067/MODIS/MOD08_D3.006

Page 16, line 479: I don't understand the argument that R > 0.6 for OMAERUV is unrealistic. The ALH in OMAERUV is mainly from CALIOP climatology, or a CTM; the UVAI is only used for selecting aerosol type. And it is not clear how much help the screening with AERONET data helps for a better correlation; if this is the reason to not

exclude the OMAERUV data, then just don't apply the screening.

**Author's response:**

The OMAERUV ALH is partly depended on UVAI. According to Torres et al. (2013), OMAERUV ALH is designed to retrieve accurate aerosol properties of absorbing aerosols in the UV channel (Torres et al., 2013). In the OMAERUV algorithm, the UVAI (together with COI) is used to determine the aerosol types. The OMAERUV ALH determination is based on the given aerosol types and other conditions (UVAI, viewing angles). For strong absorbing aerosols, the ALH is determined to be higher. For weak-absorbing or scattering aerosols, the ALH is artificially determined at lower altitude. For example, if the aerosol type is determined as sulfate, the ALH is 0 km by default. If the aerosol type is black carbon with UVAI smaller than 0.5, the ALH is 1.5 km. Thus, the UVAI dependence on ALH may be too 'optimistic' than the reality.

Please refer to the previous responses on AERONET data screen.